# DISTRIBUTIONAL REASONING IN LLMS: PARALLEL REASONING PROCESSES IN MULTI-HOP REASONING

## ABSTRACT

Large language models (LLMs) have shown an impressive ability to perform tasks believed to require "thought processes". When the model does not document an explicit thought process, it becomes difficult to understand the processes occurring within its hidden layers and determine if they can be referred to as reasoning. We introduce a novel and interpretable analysis of internal multi-hop reasoning processes in LLMs. We demonstrate that the prediction process for compositional reasoning questions can be modeled using a simple linear transformation between two semantic category spaces. We show that during inference, the middle layers of the network generate highly interpretable embeddings that represent a set of potential intermediate answers for the multi-hop question. We use statistical analyses to show that a corresponding subset of tokens is activated in the model's output, implying the existence of parallel reasoning paths. These observations hold even when the model lacks the necessary knowledge to solve the task. Our findings can help uncover the strategies that LLMs use to solve reasoning tasks, offering insights into the types of thought processes that can emerge from artificial intelligence. Finally, we discuss the implications of cognitive modeling for these results.

## 1 INTRODUCTION

The spread of activation theory in cognitive psychology suggests that ideas and concepts are stored in a network of interconnected nodes in the brain (Collins and Loftus, 1975). When one node is activated through perception, memory, or thought, it triggers a cascade that activates related nodes, facilitating processes like memory retrieval (Anderson, 1983) and association generation (Kenett et al., 2017). This theory has been instrumental in understanding how people recall information and connect different concepts, influencing cognitive research and practical applications like semantic search algorithms (McNamara, 1992; Hahn and Chater, 1998; Hofmann et al., 2011). An alternative approach In cognitive psychology is the propositional approach (Johnson-Laird, 1983). It contrasts sharply with the associative approach by focusing on the logical structure and truth values of beliefs and judgments rather than mere connections between ideas. Propositional reasoning concerns how individuals assess, validate, and infer relationships between different propositions, considering their truthfulness and logical consistency. This method involves a more deliberate and conscious level of thought, requiring the cognitive system to engage in analysis and critical thinking. On the other hand, the associative approach operates on automatic processes, where thoughts and memories are triggered by simple connections or links between ideas without evaluating their truth value (Holyoak and Morrison, 2005; Oaksford and Chater, 2007; Sperber and Noveck, 2004; Elqayam and Evans, 2011; De Neys and Bonnefon, 2013; Pennycook et al., 2015a;b). This results in a more instinctual and less reflective form of cognition, demonstrating how both approaches play distinct roles in human thought and understanding.

In the field of artificial intelligence, large language models (LLMs) have demonstrated a remarkable capability to complete tasks believed to require "thought processes" (Wei et al., 2022; Bubeck et al., 2023; Achiam et al., 2023). Originating from cognitive psychology, this notion of a thought process hinges on the ability to manipulate information in an abstract space, commonly referred to as *working memory* (Miyake and Shah, 1999; Baddeley, 2003). For example, consider the question:

"What is the first letter of the name of the color of a common banana?". Did you say to yourself or imagine the word "yellow" when trying to answer? The average response will be "yes". The chain-of-thoughts (CoT) method (Wei et al., 2022) has been the most recent success story for LLMs in solving tasks that require holding an intermediate state. This method involves LLMs noting subtask answers, eventually leading to the final answer. This approach resembles the propositional reasoning approach, and LLMs will likely adopt this strategy to generate human-like text.

However, Unlike the case of CoT, which encourages the model to mimic a human-like thought process, the training process underlying LLMs imposes no constraints on the internal process that generates the output. Thus, when not writing down an explicit thought process, the model could adopt various strategies to solve multi-hop tasks (Figure 1). This raises an important question: what strategy does the model use when applying the implicit approach? Recent studies have investigated the mechanisms that enable models to directly answer multi-hop questions (i.e., through a single token prediction). Yang et al. (2024) showed that during inference, the embeddings at the position of the bridge entity's descriptive mention offer a higher probability for the intermediate result than prompts that do not refer to this entity. Li et al. (2024) investigated the root causes of failures in directly answering compositional questions. Their findings revealed that successful prompt examples showed an increased probability of intermediate results in the middle layers. Both studies demonstrated through interference experiments that modifying the embeddings to increase the probability of the intermediate answer also affected the final answer.

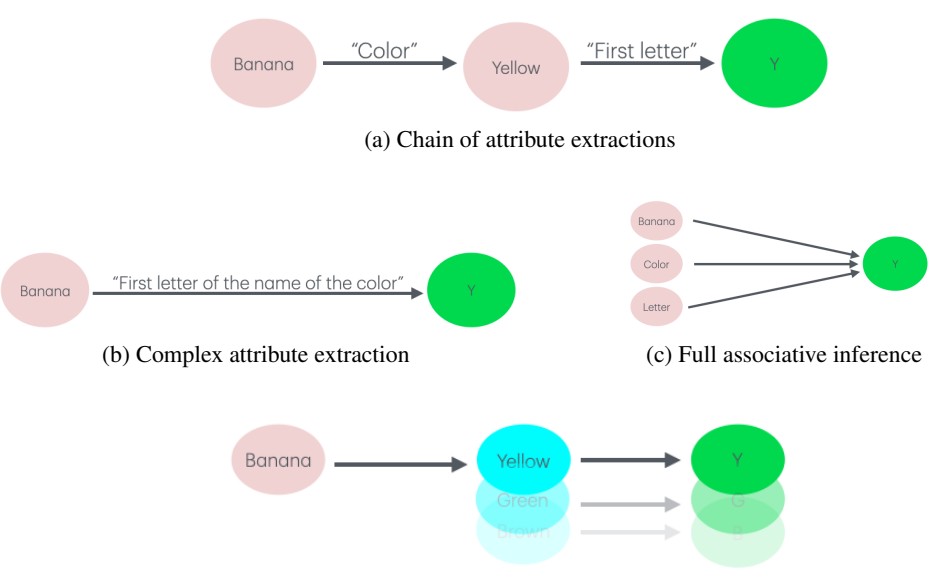

(a) Chain of attribute extractions

(b) Complex attribute extraction      (c) Full associative inference

(d) Distributional reasoning

Figure 1: Illustration of possible strategies to answer the question: *What is the first letter of the name of the color of a common banana?*: **(a)** The extraction of the *color* attribute creates a bridge entity from which the second attribute will be extracted; **(b)** Only a single extraction of the specific attribute, *first letter of the name of the color*, is performed; **(c)** The words *banana*, *color*, and *letter* are statistically related to the output *y*; **(d)** The extraction of the *color* attribute results in a distribution of bridge entities. From these entities, the second attribute will be extracted.

This work focuses on compositional two-hop questions. These can be formalized as a sequence of two attribute extractions (e.g., *What is the first letter of the name of the color of a common banana?*); the second extraction relies on information from the first (*y* from *yellow*). The main findings of this work suggest that the middle layers of LLMs not only represent the results of the first attribute extraction (i.e., *yellow*) but also this phenomenon is distributed over the range of possibilities (i.e., *yellow, brown, green*). We propose that the first attribute extraction creates a distribution of possible attributes while the second extraction operates on this distribution simultaneously (Figure 1d). This concept resembles the spread of activation theory.

Our proposal, which we refer to as a **distributional reasoning**, is demonstrated by showing that activations of potential final answers in the output layer can be approximated using a linear model which operates on the potential intermediate answers from the middle layers. We also show that, after the middle layer of the network, the inference process of compositional reasoning questions is characterized by highly interpretable hidden embeddings, which can be divided into two phases: (1) Increasing the activation of potential intermediate answers and (2) reducing the activation of intermediate answers while enhancing potential final answers (example in Figure 2a). The majority of this phase transition is handled by the feed-forward blocks (see Appendix A). Without testing direct causality, we demonstrate a strong relation between the distributions of intermediate answers and their corresponding final answers (example in Figure 2b). Lastly, we conduct two experiments that show that LLMs use the same reasoning process even when they hallucinate their answers. By forcing the models to solve reasoning tasks which are based on fictitious items, we can assess better how generalized their reasoning abilities are. This approach offers a novel method for creating datasets for evaluating internal processes of LLMs.

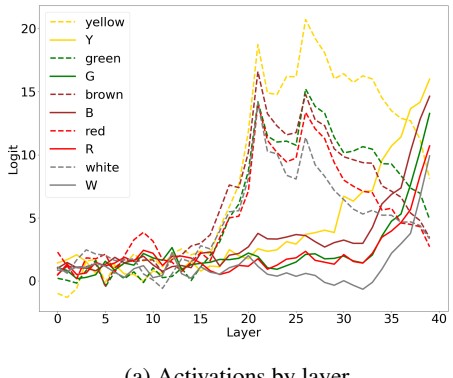

(a) Activations by layer

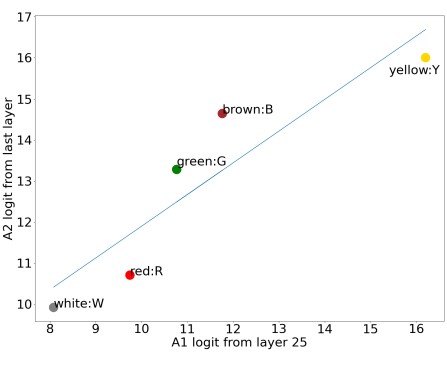

(b) Correlation between categories

Figure 2: An example of distributional reasoning in Llama-2-13B using the prompt "*What is the first letter of the name of the color of a common banana? The first letter is* ". We projected the embeddings from the hidden layers into the vocabulary space and analyzed the activation pattern of the intermediate and final answers. **(a)** The dashed lines represents activations of intermediate answers $\vec{A1}$ (color names), while the solid lines represent the activations of final answers $\vec{A2}$ (letters) by layer. A phase transition in the activation patterns is observed around layer 30. **(b)** Activations of intermediate answers $\vec{A1}$(colors) extracted from layer 25 (x-axis), compared to activations of final answers $\vec{A2}$ (letters) extracted from the last layer (y-axis).

The described reasoning process emerges from the end-to-end training approach of LLMs, which aims to optimize model output without additional constraints beyond its architecture. This characteristic makes LLMs particularly intriguing as models for providing valuable insights into cognitive modeling. This reasoning process consists of associative-like activations, which gradually and simultaneously activate semantic ideas related to the task but not necessarily essential for solving it. When observing the complete process, this associative-like pattern constructs a structured, propositional-like reasoning process, which consists of distinct stages (reasoning hops). By demonstrating that LLMs utilize both approaches in their operations, the paper not only sheds light on the internal workings of these models but also provides a computational model that mirrors these two major cognitive approaches. This helps bridge the understanding between human cognitive processes and artificial reasoning mechanisms, contributing valuable insights to the ongoing debate on how cognition can be modeled and replicated in machines.

**Contributions:**

- Novel and interpretable analysis of the multi-hop reasoning process that considers parallel and alternative reasoning paths.

- Statistical analysis demonstrating that the reasoning outcome of the model can be approximated by applying a simple linear transformation to a small and interpretable subset of logits from the middle layers. This method reveals the extent to which the second-hop operation is invariant to the prompt specifics.

- New dataset of fabricated items, allowing us to trace the reasoning process while decoupling it from the content stored in LLMs. This approach provides a novel method for creating datasets to assess internal processes in LLMs.

- Computational framework demonstrating the role of associations in structured reasoning.

Section 2 discusses related work, which includes reasoning in LLMs and approaches for interpretability. Section 3 defines crucial notations, describes our model for distributional reasoning, and provides details about the dataset we used. Section 4 presents our experiments that demonstrate the phenomenon of distributional reasoning, along with the detailed results. Section 5 discusses the implications of our results, including potential future directions, and Section 6 presents the limitations of our work.

## 2 RELATED WORK

**Reasoning in LLMs.** An established line of work has attempted to assess and enhance the capability of LLMs to solve complex tasks (Wei et al., 2022; Press et al., 2023). Most recent successes were achieved thanks to the methods of Chain-of-Thoughts (Wei et al., 2022), which involves LLMs noting subtask answers. Others addressed the ability of LLMs to manage the entire reasoning process in its hidden layers and answer in a single token prediction (Sakarvadia et al., 2023; Yang et al., 2024; Li et al., 2024).

**Interpretability in LLMs.** Many studies have attempted to interpret the internal processes occurring in LLMs during prediction (Vig et al., 2020; Geiger et al., 2021; Wu et al., 2024). This included identifying the roles of various modules in the model (Elhage et al., 2021; Geva et al., 2023; Gat et al., 2023; Li et al., 2024) and developing methods for verbally describing how the output prediction is constructed (nostalgebraist, 2020; Geva et al., 2022; Chen et al., 2024). Other studies suggest that semantic relations in LLMs are represented as linear relations (Gurnee and Tegmark, 2023; Park et al., 2023; 2024), and some of the layers' operations can be approximated by applying linear mappings (Din et al., 2023). This paper contributes to the collective effort to understand the prediction processes in LLMs and simplify them by approximating them as linear operations.

## 3 BACKGROUND

### 3.1 NOTATION

In line with the notation used by Press et al. (2023), every two-hop compositional reasoning question can be formulated using five variables: **Subject** - the initial topic the question is about; **Q1** - the first hop question that extracts an attribute from the subject; **A1** - the answer for Q1; **Q2** - the second hop question that extracts an attribute from A1; **A2** - the answer for Q2, which should be the final answer to the entire question. Table 1 presents a concrete example of this formulation. In addition, this paper will use several more notations as follows: **Category** - This refers to a semantic group of attribute names (e.g., colors, letters, cities, etc.). **Representative Token** - This is a single token from the model vocabulary associated with a specific word or expression (e.g., "US" for "The United States", "P" for "Pound", etc.). At times, the term representative token may be shortened to "token".

To analyze the extent to which a term is represented in a single embedding vector, we can utilize the LM head in a technique commonly known as the Logit Lens (nostalgebraist, 2020). The LM head is the matrix that the model uses to project the output of the final layer into a vector in the vocabulary space. We will use the term **activation** of a word in a specific layer to refer to the result of activating the LM head on the output of this layer and selecting the index of the representative token of this word from the result. Formally:

$$activation_l(word) = (Wx^l)_t$$

Table 1: Compositional reasoning notation. To illustrate the notation, we use our running example.

| Notation | Example |
| --- | --- |
| Question | What is the first letter of the name of the color of a common banana? |
| Subject | Banana |
| Q1 | Color of (Banana) |
| A1 | Yellow |
| Q2 | First letter of (Yellow) |
| A2 | Y |

Where $x^l$ is the normalized output of layer number $l$, $W$ is the LM head, and $t$ is the index of the representative token of the word. This technique is widely used to extract semantic interpretations from hidden embeddings (Geva et al., 2023; Yang et al., 2024; Li et al., 2024). We use the terms **activation** or **logit** interchangeably.

Lastly, we use the term **activation vector of category**, denoted as $\vec{A1}$ or $\vec{A2}$, to refer to the activations of an entire category.

### 3.2 LINEAR APPROXIMATION OF DISTRIBUTIONAL REASONING

We aim to define the two-hop reasoning process in two stages: (1) from a prompt to an activation vector of the intermediate answers category ($\vec{A1}$), and (2) a transformation from this activation vector to the final activation vector ($\vec{A2}$). Stage (1) is operated by a function that extracts potential attributes from a given subject. We hypothesize in this work that Stage (2) can be modeled using a linear transformation between the two category spaces. According to our formulation there is a matrix $Q2$ that, given a subject and a function $f_{Q1}$, can approximate the final vector $\vec{A2}$ as follows:

$$\vec{A1} \in \mathbb{R}^{c_1}, \vec{A2} \in \mathbb{R}^{c_2}, Q_2 \in \mathbb{R}^{c_2 \times c_1}$$

$$\vec{A1} = f_{Q1}\left(subject\right)$$

$$\vec{A2} = Q_2 \times \vec{A1}$$

The variables $c_1$ and $c_2$ represent the sizes of the semantic categories of the intermediate and final answers, respectively. Most importantly, the $Q_2$ matrix is invariant to the subject, as it is defined solely by the second-hop question.

### 3.3 DATASETS

All experiments conducted in our study are based on the Compositional Celebrities dataset presented by (Press et al., 2023). We use $6,547$ prompts divided into $14$ question types for our models and analyses. Each question pertains to an attribute of a celebrity's birthplace. For the semantic category of $\vec{A1}$ we used all of the $117$ countries used as intermediate answers in the dataset. For each of the $14$ question types, the semantic category of $\vec{A2}$ is defined by all the final answers associated with that type. Regarding the representative tokens, for each word or term, we generally use the first token capable of completing the input prompt with that term. Additionally, the prompts were modified so that the next likely token would directly answer the two-hop question (see Appendix B.1). This was done to ensure that the model will attempt to predict relevant tokens for our experiment (i.e., tokens from $\vec{A2}$). Full details can be found in our codebase, which we include as part of our supplementary material.

#### 3.3.1 HALLUCINATIONS DATASET

We introduce a unique dataset, based on the Compositional Celebrities dataset. This dataset is distinctive because it contains two-hop questions that do not have correct answers. It was designed to encourage the model to "hallucinate" potential answers and perform manipulations on them. It divided into two sets: The first set contains 1400 questions in the same format of the questions in

the Compositional Celebrities dataset, but all questions are regarding fictitious persons (see name list in Appendix B.2). The second set contains 3 question types: "What is the color of the favorite fruit of <name>? The name of the color is", "What is the first letter of the name of the favorite fruit of <name>? The first letter is" and "What is the first letter of the name of the favorite vegetable of <name>? The first letter is". Full details can be found in our codebase, which we include as part of our supplementary material.

# 4 EXPERIMENTS AND RESULTS

In this section we display our main results. All the experiments mentioned were conducted using the open-source LLMs Llama-2 (Touvron et al., 2023) with size 7B and 13B, Llama-3 (AI@Meta, 2024) with size 8B, and Mistral (Jiang et al., 2023) with size 7B.

## 4.1 LINEAR TRANSFORMATION BETWEEN TOKEN CATEGORIES

To test our hypothesis regarding the existence of the Q2 matrix (see Section 3.2), we construct a linear model for each of the $14$ question types, following the same steps. We begin by extracting the logits of $\vec{A1}$ from every layer during the inference process. For each layer, we attempt to predict the logits of $\vec{A2}$ in the final layer by using a linear regression model coupled with the k-fold method ($k = 5$). We fitted a linear model for each of the 14 categories, predicting all $\vec{A2}$ logits simultaneously. We then calculated $R^2$ between the predictions and true values for each of the $\vec{A2}$ logits predictions. For each category, we calculated the mean $R^2$ by averaging the individual $R^2$ values for each $\vec{A2}$ logit. The reported $R^2$ per category is this computed mean. Figure 3a presents an example of one regression model results, and the mean $R^2$ across all categories is presented in Figure 3b. Detailed results by LLM and category are presented in Appendix C.1.

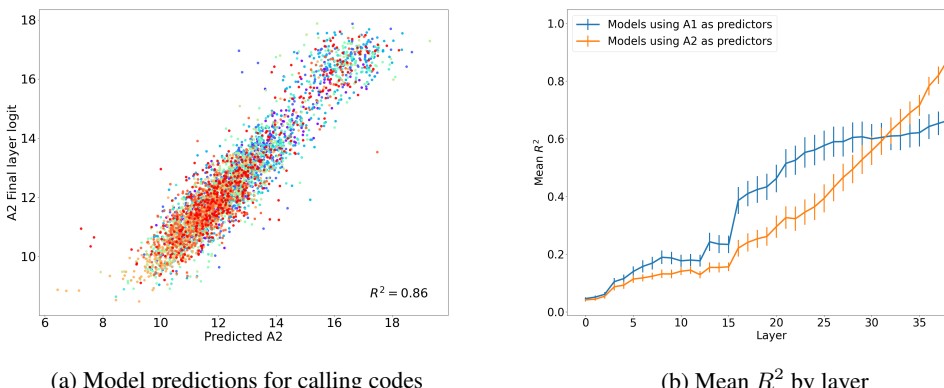

(a) Model predictions for calling codes    (b) Mean $R^2$ by layer

Figure 3: Tokens of the intermediate answers $\vec{A1}$ can approximate the tokens of the final answers $\vec{A2}$ using a linear transformation. We fitted regression models using k-fold (k=5) method to predict $\vec{A2}$ from $\vec{A1}$. Results using Llama-2-13B: **(a)** Our model predictions for question type "calling-code". This model predicts the the activation of possible first digits (1-9) using the activation of 117 countries from layer 25. x-axis - $\vec{A2}$ predicted activations; y-axis - real $\vec{A2}$ activations. Each color represent another digit (mean $R^2 = 0.86$). **(b)** Mean $R^2$ (with error bars denoting standard deviations normalized by the squared root of the group size) of our model across 14 question types, calculated for each layer separately. In blue - mean $R^2$ of the models using the logits of $\vec{A1}$ as predictors. In orange - mean $R^2$ of the models using the logits of $\vec{A2}$ as predictors. On average, the intermediate category $\vec{A1}$ was more informative about the final answers.

The results show that once two-thirds of the model depth is reached, the activations of $\vec{A1}$ can linearly predict the activations of $\vec{A2}$ in the final layer, with a mean of $R^2 > 0.5$ across various

models and question types. We interpret this observation as evidence of the strong association that occurs in LLMs between intermediate and final results in compositional reasoning.

In the next step, we repeat the same modeling method with a minor modification. This time, we attempt to predict the $\vec{A2}$ logits in the final layer using the same $\vec{A2}$ logits from each of the other layers. The results suggest that, on average across all question types, the logits from the mid-layers of $\vec{A1}$ provide more information about $\vec{A2}$ than the logits of $\vec{A2}$ themselves (Figure 3b). This again, supports the role of the intermediate answers in the forming of the final answers generated by the LLMs.

### 4.2 INTERPRETABLE REPRESENTATION OF THE INTERMEDIATE CATEGORY

We continue by examining the dynamics of the activations of $\vec{A1}$ and $\vec{A2}$. Our analysis indicates that after the middle layers of the network, there is an increase in the activation of multiple tokens from $\vec{A1}$ (Figure 4a). On average, the embeddings from the mid-layers assign a high probability to the most relevant token of $\vec{A1}$, sometimes even making it the most probable next token, even though this token is unsuitable for continuing a coherent sentence. In the subsequent layers, a phase transition occurs where the tokens of $\vec{A1}$ decrease as the tokens of $\vec{A2}$ increase, continuing this trend until the output is generated (Figure 4a).

Interestingly, there seems to be a connection between the activation patterns of the two categories in terms of the order of the activations (Figure 4b). The activation patterns of all tested LLMs are displayed in Appendix C.3. To investigate the relationship between the two activation patterns, we created a new vector, $\vec{S1}$, by sorting the logits of $\vec{A1}$ in decreasing order of their activation. We then created the following $\vec{S2}$ vector: for every index $i$ in $\vec{S1}$, the value of $S2_i$ corresponds to the activation of the $\vec{A2}$ logit of the correct final answer that matches the representative token of $S1_i$. For example, in the banana-color question, if the sorted $\vec{S1}$ contains the activations of $[yellow, brown, green]$, the respective $\vec{S2}$ will contain the activations of $[y, b, g]$. We calculated the average of $\vec{S1}$ and $\vec{S2}$ across the entire dataset (6547 prompts) and selected the top 10 logits from each vector. As a result, we obtained a vector representing the average of the top 10 logits for $\vec{A1}$ and another vector of $\vec{A2}$ logits that correspond to these top 10 A1 logits. To study the correlation between the activation patterns of $\vec{S1}$ and $\vec{S2}$, we calculated the Spearman correlation between them. The mean results are presented in Figure 4c, and category-level results are detailed in Appendix C.3. The results indicate that, on average, once two-thirds of the model depth is reached, the most activated logits of $\vec{A1}$ are arranged in a pattern closely related to the order of the $\vec{A2}$ logits in the output layer.

These observations are important in terms of interpretability. The increase in the activations of $\vec{A1}$ provides a lens to examine the process that led the model to its answer. This can assist in verifying the validity of thought processes and in explaining hallucinations when the response is incorrect. In addition, it raises questions about the causality of the process. returning to the banana question: if the model strongly associates the activation of *yellow* with *y*, one could argue that the activations are independent, and only exist because both tokens are attributes of *banana*. In contrast, if the model activates *yellow, brown, green*, and subsequently activates *y, b, g* in the same order, it becomes more challenging to argue that the activations are independent.

### 4.3 HALLUCINATIONS EXPERIMENTS

To further test our formulation and dissociate the operations of the Q2 matrix from the model's knowledge about the subject, we created two datasets of compositional questions based on the compositional celebrities dataset. We conducted two different experiments designed to make the model answer questions beyond its knowledge. This method is useful for demonstrating that the model uses valid reasoning processes, regardless of whether it can provide a correct answer.

#### 4.3.1 FICTITIOUS SUBJECTS

To test the consistency of Q2, we generated a list of 100 fictitious names (see Appendix B.2). We then expanded each of the 14 question types with 100 prompts related to these fictitious names.

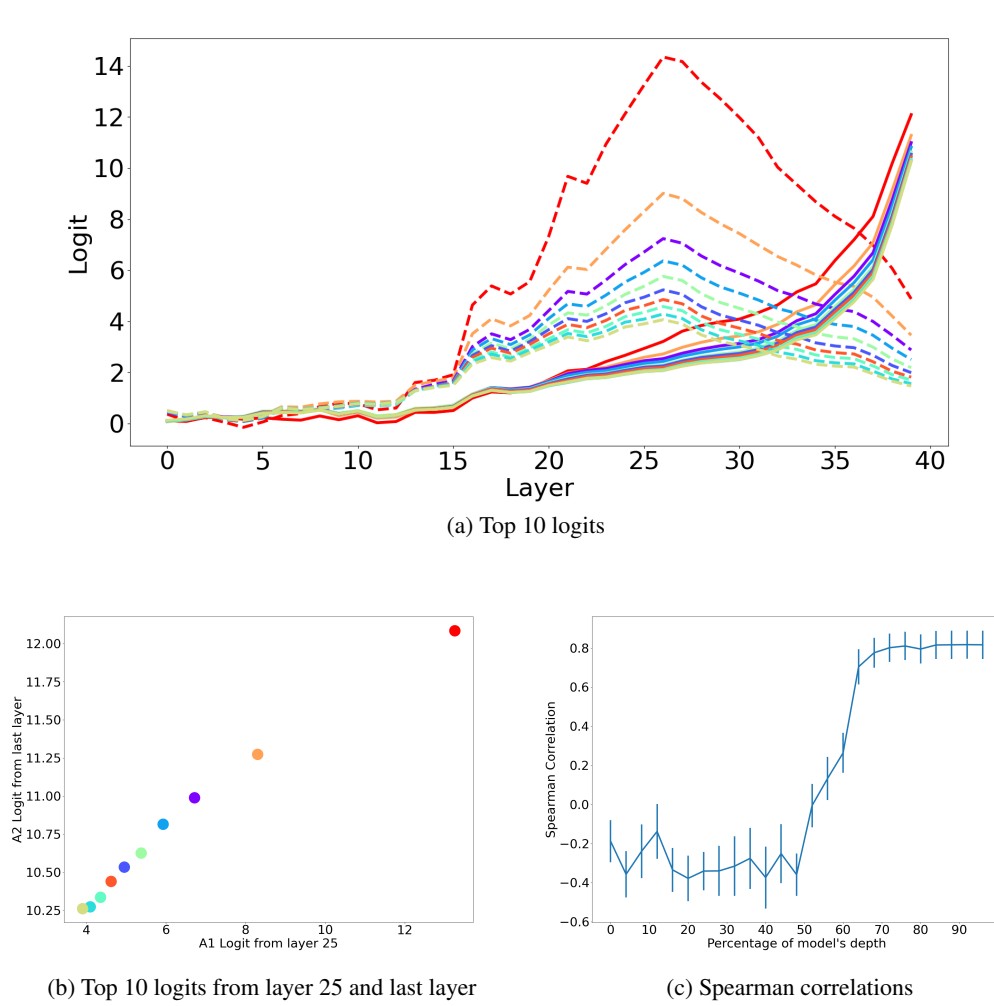

(a) Top 10 logits

(b) Top 10 logits from layer 25 and last layer

(c) Spearman correlations

Figure 4: There is a high correlation between the activation patterns of $\vec{A}1$ and $\vec{A}2$. Results of Llama-2-13B on the entire dataset: **(a)** The embeddings from the middle layers primarily represent $\vec{A}1$ (dashed lines). Then, a phase transition occurs, and the embeddings from the final layers primarily represent the $\vec{A}2$ logits (solid lines). The colors indicate pairs of intermediate answers (country names), and their corresponding correct final answers (e.g., capitals). **(b)** Both categories are sorted identically: The x-axis displays $\vec{A}1$ activations from layer 25, while the y-axis shows $\vec{A}2$ activations from the final layer. The colors indicate the same pairs from (a). **(c)** Mean spearman correlations (with error bars denoting standard deviations normalized by the squared root of the group size) across 14 question types by model depth.

We used the new prompts to evaluate our models using the following method: Initially, we fitted a linear model with Ridge regularization on the original prompts from the dataset. Then, we attempted to predict the A2 activations of the new prompts without additional training. An example of such generalization result is presented in Figure 5a, and the mean of $R^2$ by layer is presented in Figure 5b. All other experimental results are detailed in Appendix C.2. Even though the predictions are less accurate, the statistical connections derived from the dataset remain informative, even for fictitious subjects (mean $R^2 > 0.3$). The results suggest that the reasoning process is independent of the model's training data. Linear models trained on the original dataset were able to generalize to prompts about fictitious subjects. This indicates that the same reasoning process occurs within the model, regardless of the subject.

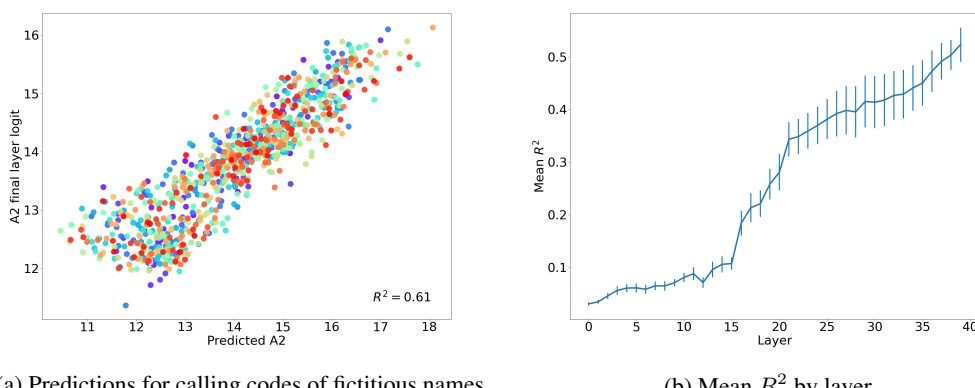

(a) Predictions for calling codes of fictitious names      (b) Mean $R^2$ by layer

Figure 5: Fictitious subjects experiment. We show that the reasoning process is dissociated from the model's training data. Our linear models generalize to prompts about fictitious subjects, indicating that the same reasoning process occurs within the model, regardless of the subject. We used the Ridge regularization method to fit linear models on the original dataset. We then tested these models on modified questions about fictitious celebrity names. Results using Llama-2-13B: **(a)** Our model generalization results (layer 25) on question type "callingcode" (mean $R^2 = 0.61$). **(b)** Mean $R^2$ (with error bars denoting standard deviations normalized by the squared root of the group size) of the fictitious subjects experiments across 14 question types, calculated for each layer separately.

### 4.3.2 FICTITIOUS ATTRIBUTES

We used 1000 person names from the Compositional Celebrities dataset and generated new two-hop question types related to unusual attributes of the subjects (e.g., their favorite fruit, see Section 3.3.1). Assuming that information regarding favorite fruits is less likely to appear in the dataset, this allows us to test whether the reasoning process remains valid under out of distribution question domains. We repeated the same modeling method described in Section 4.1, and selected results are presented in Figure 6. All other experimental results are detailed in Appendix C.2. The results suggest that distributional reasoning process exists in out-of-distribution domains as well.

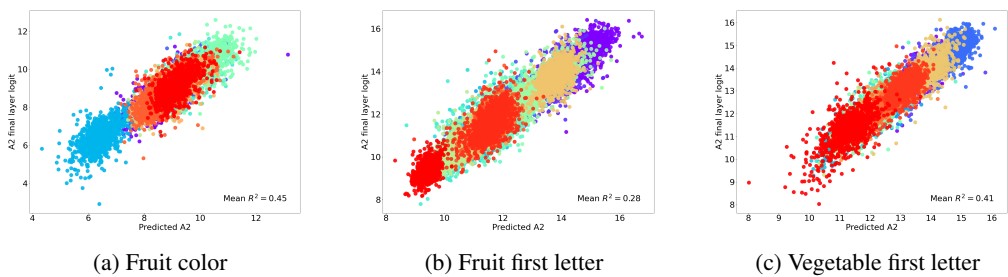

(a) Fruit color      (b) Fruit first letter      (c) Vegetable first letter

Figure 6: Fictitious attributes experiment. We observe distributional reasoning in out-of-distribution domains as well. A linear model was used to predict $\vec{A2}$ from $\vec{A1}$ on question prompts related to unusual subject attributes. Results using Llama-2-13B: **(a)** Predictions for 1000 question prompts regarding the color of celebrities' favorite fruits (mean $R^2 = 0.45$); **(b)** Predictions for 1000 question prompts regarding the first letter of celebrities' favorite fruits (mean $R^2 = 0.28$); **(c)** Predictions for 1000 question prompts regarding the first letter of celebrities' favorite vegetables (mean $R^2 = 0.41$).

## 5 DISCUSSION

This paper presents evidence of distributional reasoning in multi-hop question tasks, providing insights into the types of thought processes that can emerge from artificial intelligence. We demonstrated that by selecting a subset of tokens representing a semantic category of intermediate results, the tokens of potential final results can be approximated using a simple linear transformation. Our findings indicate that, on average, the intermediate results can explain at least 50% of the variance in the final activation results. Additionally, we demonstrated that during inference, the network's middle layers activate a small subset of tokens representing potential intermediate answers. This subset corresponds to another small subset activated in the output layers, representing potential final answers. This observation implies the presence of parallel reasoning paths, which are highly interpretable. Finally, through two dedicated experiments, we demonstrated that LLMs can manipulate information in a valid reasoning process, even when the information is hallucinated. The dynamic we capture, where the intermediate answers seem to be significant in the forming of the final answers, offers a novel cognitive approach for modeling together association and explicit reasoning. This bridges a gap that was observed by cognitive sciences decades ago and emphasis the role of AI research in cognitive modeling.

Our research, focused on observational objectives, investigates the fundamental aspects of intelligence in LLMs. However, we believe our findings can offer some practical implications. The linear approximation shown in this work is valuable not only for its computational efficiency but also for illustrating the consistency of the reasoning process, which can be viewed as a linear projection of the intermediate concepts in the semantic space. Evaluating this consistency can help assess the ability of LLMs to use valid reasoning processes, which can sometimes be more important than the output itself. For certain machine learning tasks, achieving accuracy is the primary goal, and the spurious correlations that contribute to this accuracy are not a concern. However, if our objective is to develop general human-like intelligence, it is essential to create machines with traceable and trustworthy thinking processes that can be applied to various reasoning domains.

Additionally, by tracking the activation of relevant intermediate concepts and their relations to the outcome, one can assess the validity of the answer and distinguish it from hallucination. Our findings help identify the root causes of the model's hallucinations when answering compositional questions. This is possible because the solving process is interpretable, and false associations can be tracked. Moreover, our findings show that a linear approximation can bypass half of the network depth, which should be examined in the context of early exit mechanisms (Schuster et al., 2022). We show that in multi-hop tasks, the middle layers activate tokens that are irrelevant for completing a coherent sentence (e.g., color name instead of letter), and their high probability may cause naive early exit methods to fail. However, we also show that it may be sufficient to observe the activation of the first-hop category and perform a simple manipulation to avoid unnecessary computations.

## 6 LIMITATIONS

There are a several limitations concerning the presented results. First, despite the variety in the types of questions we use, they have a similar general structure. Altering this structure could lead to different results. Second, it is worth noting that different prompt structures, question types, or subjects could lead the model to employ various solving strategies (see Figure 1). The statistical nature of the learning process likely encourages the model to utilize a variety of strategies and combine them when solving two-hop questions. Third, the proposed analysis cannot account for semantic categories that lack clear representative tokens, such as years. Future work will need to explore the mechanisms in these cases. Fourth, the results primarily rely on the Logit Lens method for semantic interpretation of hidden embeddings. While empirical evidence suggests this method can provide meaningful interpretations, it remains unclear why it works, as these LLMs were not trained for this purpose. The Logit Lens method may contain undiscovered biases and should be used with caution. Lastly, although the statistical analyses in this paper are quite convincing, they do not show direct causality. Future work will need to take this into account.

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

# APPENDIX

## A   THE REASONING PROCESS PATH

A previous study by Geva et al. (Geva et al., 2023) investigated the information flow in attribute extraction prompts. Their findings indicated that a significant part of the process in the initial layers occurs at the position of the subject prompt. This stage of processing is referred to as "subject enrichment". Following this stage, as the authors reported, the information from this process propagates to the final index. The remaining process is primarily handled in the final index, leading up to the model's output. Moreover, Li et al. (Li et al., 2024) identified critical modules for multi-hop reasoning tasks. They found that, up to the middle layers, the feed-forward blocks at the subject's position were the most significant. In the later stages, the most important modules were the multi-head attention blocks and the feed-forward at the final index.

In order to verify these observation on our dataset, we conducted an interference experiment for each prompt in the dataset as follows: At first, we used the model to predict the most probable token after this prompt, and saved its probability as the $baseline$ probability. Next, for each layer of the model, we input the same data into the model but interfered the prediction process. We replaced the embeddings at all positions in that layer with zeros, except for the last index. After the inferred inference, we saved the updated probability of the token from the first round. For each layer $l$, we calculate its $intervention\_score$ as follows:

$$intervention\_score^l = 1 - \frac{prob}{baseline}$$

The average $intervention\_score$ across the entire dataset is presented in Figure 7 (using Llama-2-13B model). The results show that on average, the influence of other token positions on the output probability significantly reduces after the 15th layer, reaching minimal effect from layer 25 onward. Considering our observations from Section 4.2, it appears that the increase in activation of the $\vec{A}1$ logits (as shown in the Figure 4a) corresponds to an information flow from other token positions. It also appears that the phase transition in the embeddings, where $\vec{A}1$ activations decrease as $\vec{A}2$ enhances, is managed solely at the last token index.

## B   DATASETS

### B.1   PROMPT MODIFICATIONS

To enhance the probability that the next predicted token will directly answer the two-hop question, we have added a suffix to each prompt in the compositional celebrities dataset. The specific suffixes for each category are outlined in Table 2.

### B.2   FICTITIOUS NAMES LIST

For the creation of our hallucinations dataset (see Section 3.3.1), we used Gemini (Team et al., 2024) for auto generating the following list of 100 fictitious names: *Scarlett Evans, Oliver Morgan, Eleanor Clark, Finley Cooper, Violet Gray, Carter Edwards, Alice Brooks, Samuel Parker, Willow Moore, Henry Mitchell, Isla Bennett, Leo Turner, Evelyn Carter, Wyatt Peterson, Harper Garcia, Lucas Ramirez, Luna Patel, Logan Martin, Scarlett Lopez, Aiden Sanchez, Chloe Lee, Owen Perez, Riley Daniels, Liam Davis, Nora Robinson, Caleb Wright, Hazel Young, Elijah Thompson, Aurora Jones, Ryan Lewis, Zoey Walker, Dylan Baker, Penelope Harris, Gabriel Allen, Charlotte Campbell, Nicholas Taylor, Amelia Jackson, Jackson Moore, Evelyn Garcia, Matthew Ramirez, Luna Lopez, Benjamin Daniels, Maya Bennett, Alexander Turner, Ava Davis, Ethan Johnson, Riley Brooks, William Peterson, Aurora Sanchez, Noah Lewis, Zoey Baker, Dylan Harris, Penelope Allen, Gabriel Campbell, Charlotte Taylor, Nicholas Jackson, Amelia Moore, Jackson Garcia, Evelyn Ramirez, Matthew Lopez, Luna Daniels, Benjamin Bennett, Maya Turner, Alexander Davis, Ava Johnson, Ethan Brooks, Riley Peterson, William Sanchez, Aurora Lewis, Noah Baker, Zoey Harris, Dylan Allen, Penelope Campbell, Gabriel Taylor, Charlotte Jackson, Nicholas Moore, Amelia Garcia, Jackson Ramirez, Evelyn Lopez, Matthew Daniels, Luna Bennett, Benjamin Turner, Maya Davis, Alexander Johnson, Ava Brooks, Ethan Peterson, Riley Sanchez, William Lewis, Aurora Baker, Noah*

Table 2: Prompt modifications

| Question type | Original prompt | Suffix | Comments |
|---|---|---|---|
| callingcode | What is the calling code of the birthplace of <name>? | The calling code is + | |
| tld | What is the top-level domain of the birthplace of <name>? | The top-level domain is . | |
| rounded_lng | What is the (rounded down) longitude of the birthplace of <name>? | The longitude is | ended with space or "-" depends on the country |
| rounded_lat | What is the (rounded down) latitude of the birthplace of <name>? | The latitude is | ended with space or "-" depends on the country |
| currency_short | What is the currency abbreviation in the birthplace of <name>? | The abbreviation is " | |
| currency | What is the currency in the birthplace of <name>? | The currency name is " | |
| ccn3 | What is the 3166-1 numeric code for the birthplace of <name>? | The numeric code is | ended with space |
| capital | What is the capital of the birthplace of <name>? | The capital is | |
| currency_symbol | What is the currency symbol in the birthplace of <name>? | The symbol is " | |
| rus_common_name | What is the Russian name of the birthplace of <name>? | The common name in Russian is " | |
| jpn_common_name | What is the Japanese name of the birthplace of <name>? | The common name in Japanese is " | |
| urd_common_name | What is the Urdu name of the birthplace of <name>? | The common name in Urdu is " | |
| spa_common_name | What is the Spanish name of the birthplace of <name>? | The common name in Spanish is " | |
| est_common_name | What is the Estonian name of the birthplace of <name>? | The common name in Estonian is " | |

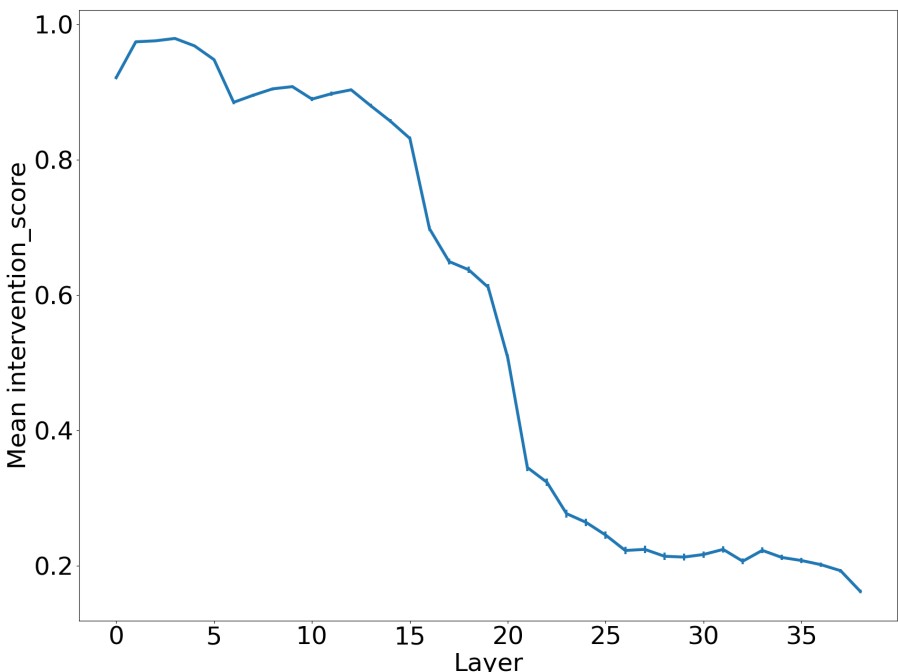

Figure 7: Intervention score using Llama-2-13B model. The average significance of any token index, except for the last one, dramatically decreases after the 15th layer.

*Harris, Zoey Allen, Dylan Campbell, Penelope Taylor, Gabriel Jackson, Charlotte Moore, Nicholas Garcia, Amelia Ramirez, Jackson Lopez, Evelyn Daniels, Matthew Bennett.*

## C  SEMANTIC TRANSFORMATIONS EXPERIMENTS

All experiments in this study were conducted using a cluster service with servers that include a single GPU and 30GB RAM, or through Google Colab services on a T4 server. The experiments were conducted using the following large language models: Llama-2-13B , Llama-2-7B, Mistral-7B (with 8-bit quantization method), and Llama-3-8B.

### C.1  MAIN RESULTS

We fitted a linear model for each of the 14 categories, predicting all $\vec{A2}$ logits simultaneously. We then calculated $R^2$ between the predictions and true values for each of the $\vec{A2}$ logits predictions. For each category, we calculated the mean $R^2$ by averaging the individual $R^2$ values for each $\vec{A2}$ logit. The reported $R^2$ per category is this computed mean. Results for each category, at two-thirds of the model's depth, can be found in Table 3. Average of Mean $R^2$ by layer can be found in Figure 8.

### C.2  HALLUCINATIONS EXPERIMENTS RESULTS

The results of the hallucinations experiments (see Section 4.3.1) are detailed by category and LLM in Table 3. The outcomes for the **fictitious subjects** experiments are shown under the *FN* columns, while the results for the **fictitious attributes** experiments appear in the bottom rows.

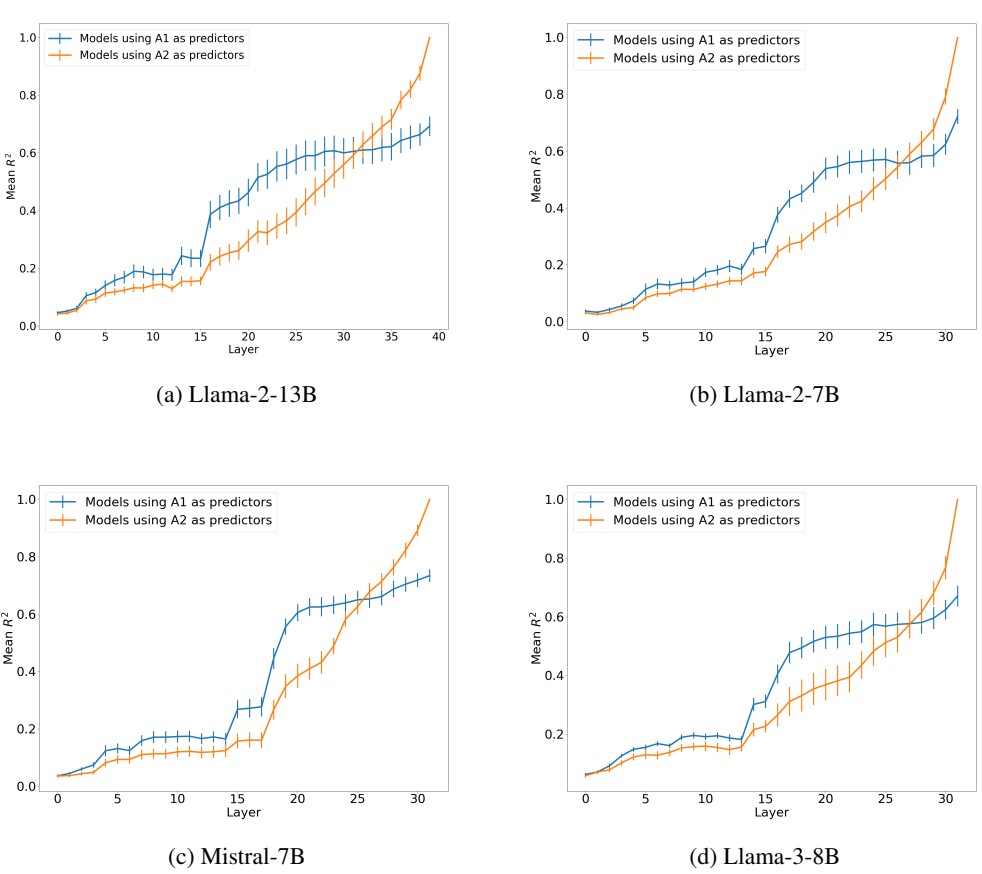

(a) Llama-2-13B

(b) Llama-2-7B

(c) Mistral-7B

(d) Llama-3-8B

Figure 8: Mean $R^2$ (with error bars denoting standard deviations normalized by the squared root of the group size) of our models across 14 question types, calculated for each layer separately. In blue - mean $R^2$ of the models using the logits of $\vec{A1}$ as predictors. In orange - mean $R^2$ of the models using the logits of $\vec{A2}$ as predictors.

Table 3: $R^2$ of linear regressions models. Columns A1 and A2 represent the categories of the semantic transformations predicted by the models; The results for the model at two-thirds depth of the LLM are displayed in the $\frac{2}{3}L$ columns; The *FN* columns show the results for the experiments involving fictitious subjects; The final divided layers correspond to the experiments with fictitious attributes.

| Transformation | | Model | | | | | | | |
|---|---|---|---|---|---|---|---|---|
| | | Llama2-13B | | Llama2-7B | | Mistral-7B | | Llama3-8B | |
| A1 | A2 | $\frac{2}{3}L$ | FN | $\frac{2}{3}L$ | FN | $\frac{2}{3}L$ | FN | $\frac{2}{3}L$ | FN |
| countries | calling codes | 0.86 | 0.61 | 0.76 | 0.47 | 0.84 | 0.68 | 0.56 | 0.4 |
| countries | domains | 0.72 | 0.42 | 0.58 | 0.45 | 0.6 | 0.41 | 0.59 | 0.29 |
| countries | longitudes | 0.54 | 0.27 | 0.61 | 0.36 | 0.67 | 0.34 | 0.54 | 0.19 |
| countries | latitudes | 0.78 | 0.37 | 0.54 | 0.18 | 0.68 | 0.46 | 0.57 | 0.26 |
| countries | currency shorts | 0.74 | 0.58 | 0.67 | 0.5 | 0.68 | 0.45 | 0.72 | 0.53 |
| countries | currency names | 0.75 | 0.47 | 0.69 | 0.45 | 0.72 | 0.44 | 0.69 | 0.46 |
| countries | iso 31661-1 | 0.52 | 0.32 | 0.64 | 0.1 | 0.5 | 0.42 | 0.58 | 0.29 |
| countries | capitals | 0.59 | 0.48 | 0.39 | 0.19 | 0.6 | 0.4 | 0.59 | 0.26 |
| countries | currency symbols | 0.78 | 0.53 | 0.68 | 0.4 | 0.71 | 0.5 | 0.78 | 0.58 |
| countries | russian names | 0.22 | 0.22 | 0.32 | 0.33 | 0.46 | 0.5 | 0.26 | 0.26 |
| countries | japanese names | 0.45 | 0.42 | 0.34 | 0.28 | 0.53 | 0.54 | 0.4 | 0.31 |
| countries | urdu names | 0.46 | 0.14 | 0.49 | 0.07 | 0.62 | 0.41 | 0.36 | 0.06 |
| countries | spanish names | 0.3 | 0.17 | 0.35 | 0.37 | 0.42 | 0.3 | 0.38 | 0.22 |
| countries | estonian names | 0.34 | 0.33 | 0.48 | 0.47 | 0.55 | 0.46 | 0.33 | 0.26 |
| fruits | colors | 0.45 | | 0.52 | | 0.33 | | 0.39 | |
| fruits | letters | 0.27 | | 0.44 | | 0.39 | | 0.42 | |
| vegetables | letters | 0.42 | | 0.46 | | 0.38 | | 0.56 | |

Table 4: Spearman correlation for average 10 top answers. The $\frac{1}{2}L$ and $\frac{2}{3}L$ columns correspond to the results at half and two-thirds of the model depth, respectively. *Note:* $^{***}p < 0.001$, $^{**}p < 0.01$, $^{*}p < 0.05$.

| | Model | | | | | | | |
|---|---|---|---|---|---|---|---|---|
| | Llama2-13B | | Llama2-7B | | Mistral-7B | | Llama3-8B | |
| Question type | $\frac{1}{2}L$ | $\frac{2}{3}L$ | $\frac{1}{2}L$ | $\frac{2}{3}L$ | $\frac{1}{2}L$ | $\frac{2}{3}L$ | $\frac{1}{2}L$ | $\frac{2}{3}L$ |
| Calling code | 0.98*** | 1.00*** | 0.38 | 0.96*** | 0.72* | 0.99*** | 0.89*** | 0.99*** |
| Domain | 1.00*** | 1.00*** | -0.85** | 0.99*** | -0.72* | 0.99*** | 0.76* | 1.00*** |
| Longitude | 0.92*** | 0.92*** | 0.94*** | 0.95*** | 0.92*** | 0.92*** | 0.49 | 0.58 |
| latitude | 0.44 | 0.44 | 0.19 | 0.24 | 0.54 | 0.54 | 0.77** | 0.77** |
| Currency short | 0.95*** | 0.95*** | 0.15 | 1.00*** | 0.64* | 1.00*** | 0.72* | 0.94*** |
| Currency name | 0.99*** | 0.99*** | -0.32 | 0.90*** | 0.5 | 0.99*** | 0.47 | 0.77** |
| ISO 3166-1 | 0.1 | 0.1 | -0.90*** | -0.96*** | 0.89*** | 0.95*** | -0.45 | -0.44 |
| Capital | 0.96*** | 0.99*** | 0.2 | 0.92*** | 0.93*** | 1.00*** | -0.09 | -0.1 |
| Currency Symbol | 0.99*** | 0.99*** | -0.77** | -0.18 | 0.12 | 0.99*** | 0.72* | 0.59 |
| Russian name | 0.95*** | 0.95*** | 0.3 | 0.98*** | 0.79** | 0.95*** | -0.68* | -0.68* |
| Japanese name | 0.94*** | 0.94*** | 0.92*** | 0.96*** | 0.49 | 0.70* | 0.85** | 0.88*** |
| Urdu name | 0.79** | 0.81** | -0.05 | -0.28 | 0.32 | 0.35 | -0.81** | -0.48 |
| Spanish name | 0.93*** | 0.93*** | 0.82** | 0.95*** | 0.42 | 0.61 | 0.95*** | 0.96*** |
| Estonian name | 0.41 | 0.44 | 0.43 | 0.96*** | -0.62 | 0.02 | 0.33 | 0.37 |

## C.3 ACTIVATION PATTERNS

Figure 9 presents the activation patterns of the top 10 $\vec{A1}$ logits and their corresponding $\vec{A2}$ logits (see Section 4.2) of each LLM. Table 4 presents the Spearman correlations of the top 10 $\vec{A1}$ logits and their corresponding $\vec{A2}$ logits sorted by LLM and layer.

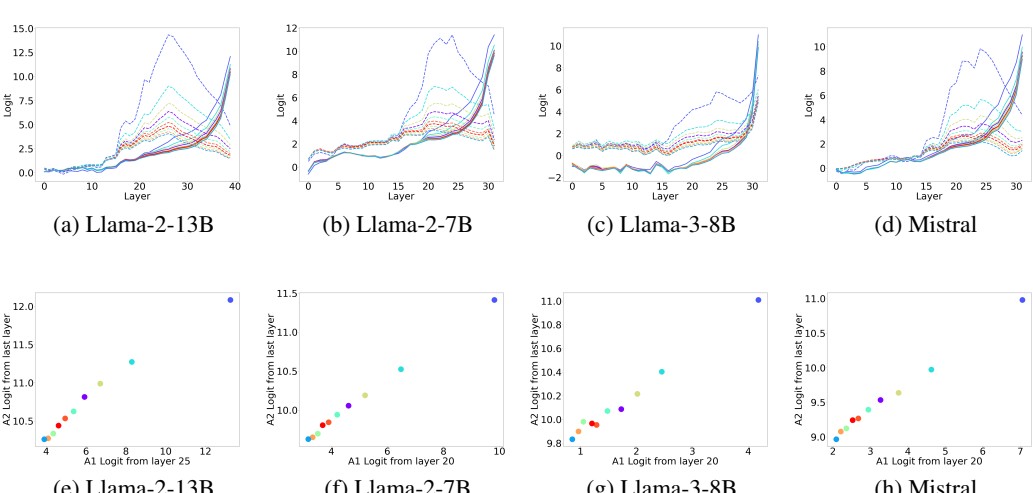

(a) Llama-2-13B  (b) Llama-2-7B  (c) Llama-3-8B  (d) Mistral

(e) Llama-2-13B  (f) Llama-2-7B  (g) Llama-3-8B  (h) Mistral

Figure 9: **(a)-(d)** The embeddings from the middle layers primarily represent $\vec{A}1$ (dashed lines). Then, a phase transition occurs, and the embeddings from the final layers primarily represent the $\vec{A}2$ logits (solid lines). The colors indicate pairs of intermediate answers (country names), and their corresponding correct final answers (e.g., capitals). **(e)-(h)** Both categories are sorted identically: The x-axis displays $\vec{A}1$ activations from the two-thirds layer, while the y-axis shows $\vec{A}2$ activations from the final layer.

Table 5: Extra relations dataset and results. The table present how many samples we used for the linear regression fit, the subjects in which the question was about, the semantic categories of A1 and A2, their sizes, and mean $R^2$ in $\frac{2}{3}$ depth of the model.

| Samples | Subject | A1 | A1 size | A2 | A2 size | Mean $R^2$ |
|---|---|---|---|---|---|---|
| 524 | sport players | sports | 8 | letters | 8 | 0.75 |
| 198 | flowers, fruits, vegetables, birds | colors | 10 | letters | 7 | 0.55 |
| 347 | random words | letters | 7 | colors | 10 | 0.86 |
| 524 | sport players | sports | 5 | numbers | 5 | 0.62 |

## C.4 ADDITIONAL RESULTS

To expand our empirical results to other domains than those presented in the Compositional Celebrities Dataset, we curated additional 1593 prompts divided into 4 new relation types which do not relate to countries or celebrities' birthplaces. The question types are: "What is the first letter of the sport that <> plays? The first letter is"; "What is the first letter of the color of <>? The first letter is "; "What is the color that starts with the same letter as "<>"? The color is " and "How many players are there in a team of the sport played by <>? The number of players is ". The prompts was generated using Gemini (Team et al., 2024). Details about the dataset and the results of the semantic transformation experiment using Llama-2-13B are presented in Table 5.

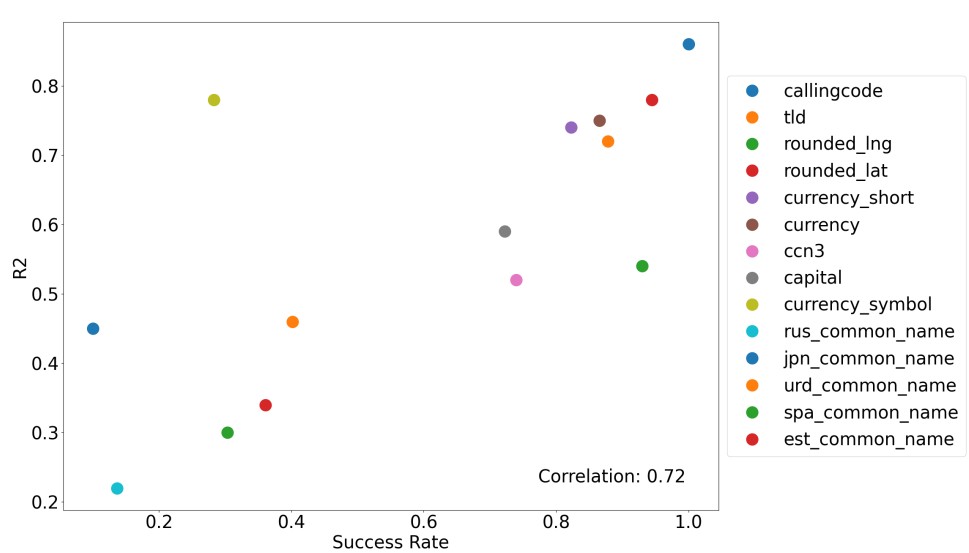

Figure 10: High correlation between the rate at which the model was able to answer questions correctly in each category and the evidence for distributional reasoning (mean $R^2$ in our regression analysis). Analysis using Llama-2-13B.

### C.5 CORRELATION TO ANSWERS CORRECTNESS

Implicit reasoning, where the model does not write down its steps, is notably more challenging for LLMs. While the Compositional Celebrities dataset was beneficial for testing our hypotheses, it was not designed for implicit reasoning, making some question types too hard for the tested models to solve in that way. When the models completely fail to reason, there might not be any reasoning process that could be followed, and therefore our regression analyses will not work.

In order to test the dominance of distributional reasoning in the LLMs' solving strategy, we conducted the following analysis using Llama-2-13B: For each prompt in the dataset, we tested whether the correct answer to the question appeared in one of the five most probable tokens predicted by the model. If it did, the prompt was classified as successful. Figure 10 shows the percentage of successful prompts in each category compared to our analyses' results (mean R² at two-thirds depth of the model). We observe a high correlation (0.72) between these values, suggesting that distributional reasoning is more dominant as a solving strategy when the model can reason effectively. Conversely, it's less dominant when the model struggles to reason, and other mechanisms are shaping the model's output.

## D TRACING REASONING PROCESS

For demonstration purposes, this section presents 4 examples of two-hop questions in which we can visually trace the reasoning process which led to the output. All four examples were generated using Llama-2-13B.

**Correct Answers** Figure 11 presents two examples of prompts where the model predicted the correct answer to the two-hop question. The visualization shows the activation of each A1 logit from the $\frac{2}{3}$ depth of the model compared to their corresponding activations of the A2 from the last layer. In both examples, we can see that the model activated the correct intermediate answer, allowing us to verify that the internal process occurred as expected.

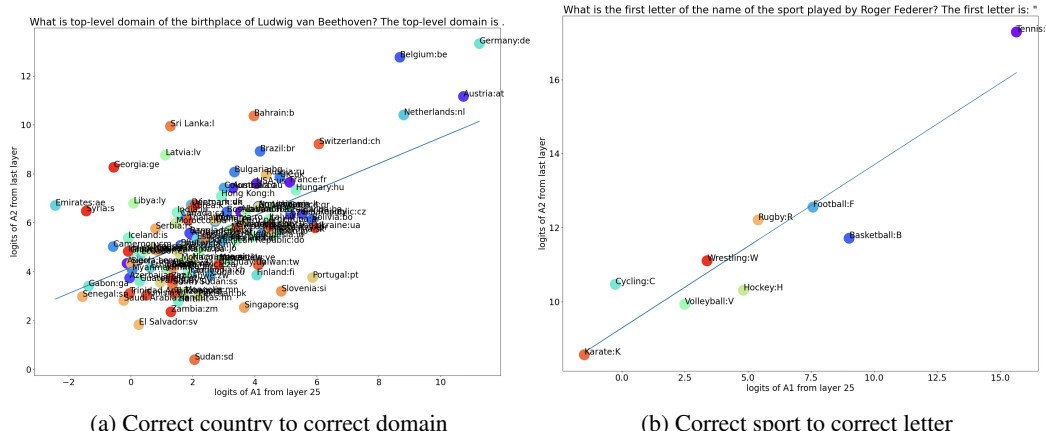

(a) Correct country to correct domain          (b) Correct sport to correct letter

Figure 11: Visualization of distributional reasoning: Two examples of prompts where the model activated the correct intermediate answers and predicted the correct output. Examples use Llama-2-13B. **(a)** Activations for the prompt: "What is top-level domain of the birthplace of Ludwig van Beethoven? The top-level domain is .". The x-axis shows the activation of different countries in layer 25, while the y-axis displays the activations of their corresponding domain names in the last layer. The model correctly activated "Germany" and accurately predicted "de". **(b)** Activations for the prompt: "What is the first letter of the name of the sport played by Roger Federer? The first letter is: ". The x-axis shows the activation of different sport names in layer 25, while the y-axis displays the activations of their corresponding letters in the last layer. The model correctly activated "Tennis" and accurately predicted "T".

**Incorrect Answers**    Figure 12 presents two examples of prompts where the model predicted an incorrect answer to the two-hop question. The visualization shows the activation of each A1 logit from the $\frac{2}{3}$ depth of the model compared to their corresponding activations of the A2 from the last layer. In both examples, we can see that the model activated the wrong intermediate answer, subsequently activated its corresponding final answer, and failed to answer the question correctly. This approach allows us to trace the source of the error in the reasoning process, which in this case stems from an incorrect answer to the first hop of the question.

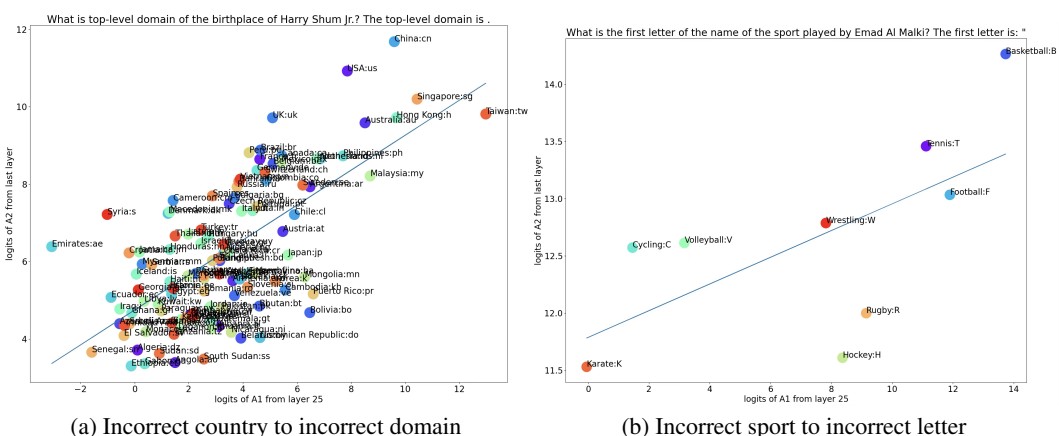

(a) Incorrect country to incorrect domain      (b) Incorrect sport to incorrect letter

Figure 12: Visualization of distributional reasoning in hallucinations: Two examples of prompts where the model activated the wrong intermediate answers and predicted the wrong output. Examples use Llama-2-13B. **(a)** Activations for the prompt: "What is top-level domain of the birthplace of Harry Shum Jr.? The top-level domain is .". The x-axis shows the activation of different countries in layer 25, while the y-axis displays the activations of their corresponding domain names in the last layer. The model incorrectly activated "China" instead of "Costa Rica", leading to an incorrect activation of "cn" rather than "cr". **(b)** Activations for the prompt: "What is the first letter of the name of the sport played by Emad Al Malki? The first letter is: ". The x-axis shows the activation of different sport names in layer 25, while the y-axis displays the activations of their corresponding letters in the last layer. The model incorrectly activated "Basketball" instead of "Karate", leading to an incorrect activation of "B" rather than "K".

