# OpenReview forum: "Distributional reasoning in LLMs: Parallel Reasoning Processes in Multi-hop Reasoning"
_ICLR.cc/2025/Conference — Submitted to ICLR 2025_

### Official Review · Reviewer_P5is · 2024-10-30

**Soundness:** 3
**Presentation:** 3
**Contribution:** 2
**Rating:** 6
**Confidence:** 3

**Summary:**

The paper investigates the inner processes of LLMs on two-hop reasoning tasks. The authors identify the categories of the intermediate and final answers and study their activation in the LLMs. They find that these categories can respectively be found in the intermediate and final layers and that the connection between the intermediate and final activations can be mapped using a linear transformation. Moreover, they find that LLMs consider multiple hypotheses in parallel, with the middle layers representing the potential intermediate answers and the final layers the potential final answers.

**Strengths:**

The paper is well written and mostly easy to read and understand.

The proposed methodology is an interesting way for studying the challenging problem of LLM interpretability. The resulting findings provide intriguing insights into the reasoning of LLMs in two-hop composition tasks and can be very useful to the research community.

**Weaknesses:**

Although the analysis provided in the paper is quite interesting, the main issue of the paper is its limitation to a single dataset and two-hop questions, which restricts the possibility to extrapolate the findings to larger settings. More specifically:

1. The activation of a word is independent from the rest of the logits, therefore this activation may not be the main reasoning mechanism happening inside the LLM but a byproduct or a minor function only marginally used in the total prediction. The paper does not clearly establish if the activation is the main reason for the final model prediction. It might be interesting to compute relative activations with respect to the other dimensions of the layer representation with (e.g. with softmax).

2. The example provided in Figure 1 is a composition of two simple queries. The category mapping found could be an artifact coming from the task instead of a property of the model on two-hop reasoning tasks. The evidence provided in the experiments does not seem to eliminate this possibility.

3. Figure 4 does not include legends for the plots, making them hard to interpret. It would be good to include what each data point/plot corresponds to.

4. The experiments focus on two-hop reasoning. To assess the robustness of the findings, it would be interesting to see if they hold to n-hop queries with n higher than two. Queries recalling names could easily be modified into three-hop by asking to return the first letter of the name instead of the full name as done in Figure 1.

**Questions:**

1. Have you conducted experiments to verify if the identified mechanism is the main function responsible for the final model prediction or how much does it contribute to it?

2. How diverse are the questions asked in the experiments? How did you assess that the dataset used contain sufficiently diverse queries for your findings to be robust?

3. Have you conducted experiment on n-hop reasoning tasks with $n>2$ to verify if your findings scale to more complex multi-hop reasoning tasks?

4. What are the average sizes of the semantic categories $c_1$ and $c_2$?

---

> ### Author Response · Authors · 2024-11-21
>
> We appreciate the reviewer's comments, and we apologize for the late response. We will address the mentioned issues to the best of our ability.
>
> Regarding the limitation about the dataset, we addressed this important point in the general comment of our rebuttal. Please refer to our answer there.
>
> W1.
>
> LLMs are statistical models, and numerous internal associations learned from their training sets can shape each prediction differently. We believe it's unrealistic to expect a single high-level mechanism to be the sole process driving model predictions.
>
> However, our analyses show that the distribution of a selected set of intermediate answers can explain a significant portion of the variance in the output distribution - this was shown while generalizing on unseen data. For instance, using the banana example from the paper, if we observe the activation of yellow, brown, and red, we can estimate the activations of "y", "b", and "r". Certainly, other associations also shape the final answer to some extent. For instance, we can surmise that the activation of "b" is slightly boosted due to its connection to the word "banana". However, our analyses demonstrate that these other associations are less significant compared to our proposed mechanism. If this were not the case, we wouldn't be able to predict the final distribution so accurately using only the selected intermediate answers.
>
> Furthermore, in cases where our model's predictions were less accurate, it appeared that the LLMs were unable to reason effectively. This also suggests that distributional reasoning plays a dominant role in the LLMs' reasoning mechanism. We've added a new section to the appendix to demonstrate this point. Please refer to section C.5 in the latest version of the paper. This analysis can provide some sense of the extent to which distributional reasoning is an essential part of the reasoning process, in addition to our main findings.
>
> If we want to delve deeper into whether the selected activations truly cause the activations in the output, we're entering the realm of causality. We agree that understanding the causes behind model predictions is crucial. We believe that developing methods for causally interpreting LLMs is essential, but it remains an open area of research [1,2,3]. While we do not claim causality, we believe that our findings are valuable regardless of whether the process has a causal nature.
>
> We also want to highlight that the output embedding, which represents the final answers, was created by performing a series of computations on the embedding from the middle layer, which represented the intermediate answers. A non-causal process means that the LLM layers manipulate the middle embedding in a way that is invariant to the coordinates holding the representations of the intermediate answers. The output of this invariant process creates a vector that is surprisingly linearly dependent on the exact coordinates, which were not part of the process. We find this alternative hypothesis unlikely, but we did not prove it to be false.
>
> Regarding your suggestion for an additional analysis (computing relative activations), we are not sure we understand your meaning. Could you elaborate on that idea?
>
> [1] Causal Abstractions of Neural Networks. https://arxiv.org/abs/2106.02997
>
> [2] CausaLM. https://arxiv.org/abs/2005.13407
>
> [3] Causal Inference in Natural Language Processing. https://arxiv.org/abs/2109.00725
>
> W2.
>
> We sincerely apologize, but we couldn't understand the reviewer's meaning on this point. Could you please elaborate? Or perhaps provide an example to illustrate?
>
> W3.
>
> We apologize for any confusion caused by our figures. We are working to clarify all of them for easier reading. We thank the reviewer for pointing out the issues with this figure, and we will address them to the best of our ability.
>
> W4.
>
> Regarding n-hop queries with more than 2 steps: We believe that our formulation can handle any number of reasoning steps by adding more linear transformations to the computation. In practice, it appears that LLMs struggle to perform more than two implicit reasoning steps (and even the two-step case is challenging). Future work might explore how to leverage our findings in order to teach models how to solve more complicated queries.

---

> > ### Author Response · Authors · 2024-11-21
> >
> > Q1. In continuation of our answer to W1, we believe our results demonstrate the dominance of the identified mechanism as they stand. We argue that the distribution of intermediate answers can explain the distribution of final answers using a constant linear function—our experiments were designed to support this specific hypothesis. Additionally, please refer to the new analysis added to section C.5 following this rebuttal.
> >
> > Q2. Currently, we have results for 21 different relations, each containing hundreds of prompts. To strengthen our findings, we calculated the results using the k-fold method, demonstrating generalization to held-out data. Furthermore, the hallucination experiments test our hypothesis in a more challenging setting, employing a novel dataset type and approach to assess internal semantic processes in a neutral manner.
> >
> > Q3. See our response to W4.
> >
> > Q4. C1 size is typically 117 (countries), while C2 sizes range from 9 (digits of calling codes category) to 117 (top-level domain category).

---

> > > ### Comment · Reviewer_P5is · 2024-11-23
> > >
> > > Thank you for your response, my questions have been satisfactorily answered.
> > >
> > > On the suggestion at the end of W1, I am referring to the unnumbered equation at the bottom of page 4. The activation function selects the logit of a single dimension $t$. I understand that only the indexes of the pre-selected token categories are visualised. It would be interesting to see if the selected tokens are the ones with the highest activation among all tokens in the vocabulary. Please, correct me if my understanding is incorrect.
> > >
> > > The comment W2 is about the robustness of the findings. As the experiments have been performed on a single dataset, the mechanisms found may be specific to this dataset and not hold on other datasets/tasks. I hope this point is clearer now. However, this has been addressed by the extra results in Section C.4.
> > >
> > > Regarding W4 and Q3, conducting experiments on n-hop queries would still be valuable, even if the performance of LLM is lower. In particular, if multi-hop reasoning in LLMs can be modelled by a sequence of linear transformations, such experiments could help better understand where and why LLMs fail on higher number of hops.
> > >
> > > I am satisfied with the authors' answers to my questions and choose to maintain my score.

---

### Official Review · Reviewer_ndbn · 2024-10-31

**Soundness:** 2
**Presentation:** 3
**Contribution:** 2
**Rating:** 3
**Confidence:** 3

**Summary:**

This paper investigates the mechanism of multi-hop reasoning of LLM. Specifically, the paper uses a compositional two-hop question answering setup as a test bed for the investigation. It proposes the hypothesis, referred to as distributional reasoning, that the first attribute extraction creates a distribution of possible attributes on which the second extraction can operate simultaneously to derive the final answer. To test the proposed hypothesis, the paper constructs a linear model to predict the activation of the final layer based on the activations of intermediate layers. The results indicate that once the model depth reaches a certain threshold, the activations of intermediate layers can linearly predict the activations of the final layer. Additionally, the paper demonstrates that the intermediate layers activate a small subset of tokens representing potential intermediate answers.

**Strengths:**

1. The paper proposes a hypothesis to interpret the multi-hop reasoning ability of LLM and designs sound experiments to test the proposed hypothesis. The findings provide interesting insights on neural cognition and reasoning.
2. The paper is clearly written. The experiments are well-designed with main results (Section 4.1 & 4.2) providing direct support to the hypothesis and additional results (Section 4.3) ruling out the confounding factor of the possibility of LLM simply memorizing knowledge.

**Weaknesses:**

1. While I can follow the proposed hypothesis and think the experiments are well-designed for testing them, I am unsure whether we can draw a conclusion that LLMs use distributional reasoning to answer multi-hop questions. As the authors also point out in the Limitations section, the paper only experiments on the two-hop question answering setup and all the questions are synthesized from one data source, the results may be different on questions with very different structures. For example, I am wondering whether the proposed hypothesis still hold on answering math problems that require multiple steps to solve. Or, still within the question answering setup, can the concept of distributional reasoning be identified on answering n-hop questions when n > 2?
2. Another major concern is the implication of the proposed hypothesis. Although the experimental results support the proposed hypothesis, it does not exclude other possibilities - therefore, I personally think the contributions of "computational framework demonstrating the role of associations in structured reasoning" is a bit oversold. A way to strengthen the contribution is to demonstrate the potential usage of the discovered hypothesis/pattern. Can the finding on distributional reasoning help us design better reasoning algorithm (e.g., discover self-inconsistency) or help us train models better at multi-hop reasoning? Even though the paper mentions some relevant aspects in Section 5, it lacks a proof-of-concept to showcase this is a promising way to go.

**Questions:**

See the questions raised in "Weaknesses".

---

> ### Author Response · Authors · 2024-11-19
>
> We appreciate the reviewer's comments and will address the mentioned issues to the best of our ability.
>
> Regarding the limitation about the dataset, we addressed this important point in the general comment of our rebuttal. Please refer to our answer there.
>
> Regarding the suggestion of using math problems: math is an interesting field that deserves its own research paper (and actually we did investigate it while working on this research project). The approach we used in this paper relies on the Logit Lens method, in which we need to assign a representative token to each term. The problem with math within this setup is that we can't really assign tokens to numbers with more than one digit, as each digit is a token by itself. We want to clarify that this doesn't mean the reasoning process doesn't occur the way we present it, but it will need a different approach to prove that in these specific cases.
>
> More specifically about math problems, take as an example the following equation: "3 + 1 + 2 =", LLMs can use different strategies to solve this problem. But most probably, it was in the training data, so there is not any reasoning process that we could trace because it is just a case of memorization.
>
> Lastly, regarding the suggestion of n-hop queries with n > 2. We believe that our formulation can handle any number of reasoning steps by adding more linear transformations to the computation. In practice, it appears that LLMs struggle to perform more than two implicit reasoning steps (and even the two-step case is challenging). We can speculate about a potential training mechanism or architecture that could increase the number of steps LLMs can perform, but we believe this would go beyond the boundaries of this paper.
>
> Regarding the second weakness mentioned by the reviewer. Our research explores the foundational elements of intelligence in LLMs. Driven by observational goals, we aim to uncover emergent properties of LLMs which show up spontaneously from the end-to-end nature of their training. This research primarily aims to enhance our understanding of the internal workings of deep learning systems.
>
> Having said that, We do believe our findings have valuable applications.
>
> First, our findings show that the reasoning process is traceable. This means we can look for intermediate answers and their alternatives to trace causes of hallucinations. Following your comment, we have added a few visual examples of this; please refer to section D in the latest revision of the paper. We will integrate this section into the main paper for the camera-ready version.
>
> Second, we believe that our proposed analyses can be used as an evaluation method for the reasoning abilities of LLMs. We believe that current benchmarks, which test only the output of the models, are missing an important notion of the internal process. We offer an evaluation method to test not only if the output was correct, but also if the process was correct.
>
> Third, we believe that the ideas presented in this paper should be integrated into future training processes of LLMs. This integration aims to teach models how to reason implicitly and, hopefully, how to reason with more than two steps. We believe that a proof of concept for this part is beyond the scope of this paper and belongs to a subsequent study. Nevertheless, this is definitely a call for using these ideas to enhance reasoning abilities.

---

> > ### Comment · Reviewer_ndbn · 2024-11-22
> >
> > I thank the authors for the detailed reply. I want to highlight that I give the Confidence score as 3 since I am not very familiar with this topic. My following questions may be too demanding but just want to point that out for reference:
> >
> > >  The approach we used in this paper relies on the Logit Lens method, in which we need to assign a representative token to each term.
> >
> > Does this mean the approach will inherit the limitation of Logit Lens method? I am still unsure whether we can draw a conclusion that LLMs use distributional reasoning to answer multi-hop questions.
> >
> > > We believe that our formulation can handle any number of reasoning steps by adding more linear transformations to the computation.
> >
> > Will this argument be too strong given the current experiment results presented in the paper?
> >
> > > Second, we believe that our proposed analyses can be used as an evaluation method for the reasoning abilities of LLMs. We believe that current benchmarks, which test only the output of the models, are missing an important notion of the internal process. We offer an evaluation method to test not only if the output was correct, but also if the process was correct.
> >
> > Shall we apply human prior to restrict how LLMs do reasoning?

---

> > > ### Author Response · Authors · 2024-11-24
> > >
> > > We appreciate the reviewer's follow-up question to our response.
> > >
> > > Q1.
> > >
> > > The Logit Lens is a popular method for interpreting hidden embeddings. Its main limitation stems from the fact that LLMs were not trained to activate logits in their hidden layers, and it remains an open question why this method works at all. However, as we observe in our paper and many others, this method yields meaningful results.
> > >
> > > While we use the Logit Lens to demonstrate our proposed mechanism, our approach does not inherently depend on it. Even if the Logit Lens method proved ineffective, distributional reasoning could still be the main mechanism of the reasoning process—we would simply need a different interpretability method (such as linear probing) to prove our claims. We want to highlight again that Logit Lens is only a method for interpreting embeddings. The process itself occurs within the embedding vectors, and the activated logits are merely (very useful) artifacts that allow us to uncover what is represented there.
> > >
> > > About the concern of the reviewer of wether we can draw conclusion that LLMs **use** our proposed mechanism. The way we see it, the term “use” in this context is referring to the causality of the process. We agree that understanding the causes behind model predictions is crucial. We believe that developing methods for causally interpreting LLMs is essential, but it remains an open area of research [1,2,3].
> > >
> > > However, while we do not claim causality, our results show that our linear model can explain a significant portion of the variance in the data, even though it was calculated on unseen data (using the k-fold method). If distributional reasoning were not a key mechanism in processing our prompts, we would be unable to predict the final distribution with such accuracy using only the selected intermediate answers. This finding highly valuable, regardless of whether the process has a causal nature.
> > >
> > > We also want to highlight that the output embedding, which represents the final answers, was created by performing a series of computations on the embedding from the middle layer, which represented the intermediate answers. A non-causal process means that the LLM layers manipulate the middle embedding in a way that is invariant to the coordinates holding the representations of the intermediate answers. The output of this invariant process creates a vector that is surprisingly linearly dependent on the exact coordinates, which were not part of the process. We find this alternative hypothesis unlikely, but we did not prove it to be false.
> > >
> > > Additionally, please refer to sections C.5 and D, which we added to the paper after this rebuttal. We believe they can provide more intuition about the dominance of our proposed mechanism in LLM reasoning.
> > >
> > > [1] Causal Abstractions of Neural Networks. https://arxiv.org/abs/2106.02997
> > >
> > > [2] CausaLM. https://arxiv.org/abs/2005.13407
> > >
> > > [3] Causal Inference in Natural Language Processing. https://arxiv.org/abs/2109.00725
> > >
> > > Q2.
> > >
> > > We meant that theoretically, adding more linear transformations would allow us to expand our formulation. The problem is that LLMs appear unable to perform more than two reasoning steps. This could go in two directions. First, future models might develop this ability through additional training data, more layers, or architectural changes—allowing us to examine how three-step reasoning works. Second, we could leverage our proposed mechanism to train models to reason using our approach. This would require developing a way to add constraints to internal representations during the training process. However, as mentioned, that experiment belongs to a future study.
> > >
> > > Q3.
> > >
> > > We thank the reviewer for this important question. The answer depends on what we aim to achieve.
> > >
> > > First, if our goal is to create human-like intelligence, then using human priors would be a sensible approach. Second, even if replicating human cognition isn't our goal, incorporating human priors could still be an effective way to enhance LLMs' reasoning capabilities (as discussed in Q2).
> > >
> > > In the quoted paragraph, we discussed an evaluation method that provides a more comprehensive view of the reasoning process. For example, if a model answers a two-step question correctly, this might stem from memorization or shortcut-learning rather than true reasoning—something our approach can detect.
> > >
> > > Moreover, our linear modeling approach—and especially our hallucination experiments—can be used to evaluate the validity of the process in a more general and pure way. If we can verify that a model can identify first letters of colors for different items in a way that proves it has no relation to the items themselves, we can be more confident in the model's ability to generalize to new tasks. Furthermore, if we consider this important, we can be more confident that we have created a model that employs reasoning processes similar to human cognition.

---

> > > > ### Comment · Reviewer_ndbn · 2024-11-27
> > > >
> > > > I really appreciate the authors for the careful response.
> > > >
> > > > I very much agree with the authors' response to Q1. I think I am just unsure "this finding highly valuable, regardless of whether the process has a causal nature" given the conclusions are drawn through limited datasets which with synthetic question setup.
> > > >
> > > > I would like to maintain my score, as I have carefully reviewed both the paper and the authors' response, yet my concerns remain unresolved. That said, I want to emphasize that **I am uncertain about the decision to accept or reject this paper, as I am not fully familiar with the standards in this field**. I kindly ask the AC to take this into consideration when evaluating my review.

---

### Official Review · Reviewer_b2q5 · 2024-11-01

**Soundness:** 3
**Presentation:** 4
**Contribution:** 3
**Rating:** 6
**Confidence:** 3

**Summary:**

This paper provides a study of the internal reasoning processes of LLMs when answering compositional questions.
They find that one can apply linear models on the activations for specific task-related tokens at intermediate transformer layers to approximate the reasoning of the model at the "outcome" final prediction layer. This supports a hypothesis that LLMs consider a distribution of possible answers at intermediate layers then narrow down the final answer at later ones, in a process akin to the parts of the "spread of activation" and propositional reasoning theories of human cognition.

The authors consider a synthetic dataset (the Compositional Celebrities dataset from Press et al 2023, as well as carefully crafted variants to test for hallucinatory scenarios) and provide strong evidence for their hypothesis. The authors argue that transformer-based LLMs can be used to test questions in cognitive modeling.

**Strengths:**

* The paper is well-written and well-motivated.
* The study design is clear and effective, and the results are interesting and well-supported. The secondary experiments (4.2 on the interpretability of the intermediate layers, and 4.3 on whether this type of reasoning still holds when the answer is hallucinatory) do a good job of preemptively answering follow-up questions I had on the original study.
* The finding will be of interest to people concerned with whether current LLMs can shed light on questions in cognitive modeling of reasoning processes.

**Weaknesses:**

**W1** The primary weakness of the paper is that it only considers one clean, synthetic dataset with one type of question. It is unclear whether such a clean result would hold on a more realistic reasoning dataset. The authors acknowledge this in their limitations section, but it is still the case that this paper shows __one__ scenario in which a clean form of reasoning can be approximated using linear modeling for __one__ kind of problem. The paper would be stronger with at least one other dataset that is more complex than 2-hop compositional questions and/or noisier than the templatic questions in Compositional Celebrities. Otherwise, this analysis should really be considered exploratory rather than comprehensive.

**W2** l501 "This bridges a gap that was observed by cognitive sciences decades ago and emphasis the role of AI
research in cognitive modeling." -- it would be helpful to describe more specifically how the result contributes the debate surrounding this gap (i.e., why they are incongruous) more specifically in the context of existing work. The paper seems to argue that both associative and propositional reasoning can take place at the same time in a single framework -- whether this is possible has been the subject of debate for many years, and various other cognitive models have been proposed that integrate both paradigms to differing degrees, e.g. [ACT-R](https://wires.onlinelibrary.wiley.com/doi/full/10.1002/wcs.1488?casa_token=9nMiql_9dRMAAAAA%3A1kKURVyse66RR4wdR3wxa1DDWQI_HO0D-oUAPfHepYuJzEhYj_0Jo77bV5qSxGHQq9O4Sz3VPpYa) and [CLARION](https://en.wikipedia.org/wiki/CLARION_(cognitive_architecture)). Comparing your consideration of transformers to these other frameworks and discussing how the contributions differ would benefit the paper.

**Questions:**

**Q1** Does this analysis generalize to any sort of task that isn't 2-step compositional reasoning? What classes of task/problem does this framework make sense for, and what kind would it be unrealistic?

**Q2** This analysis suggests that the suggest-then-narrow-down reasoning approach can be __induced__ from the model's intermediates, but that is not the same thing as the model necessarily __using__ that approach. How can we have confidence that the model is actually using this reasoning chain and not rely upon other forms of reasoning instead/ in addition?

---

> ### Author Response · Authors · 2024-11-20
>
> We appreciate the reviewer's comments, and we apologize for the late response. We will address the mentioned issues to the best of our ability.
>
> Regarding the limitation about the dataset, we addressed this important point in the general comment of our rebuttal. Please refer to our answer there.
>
>
> Regarding the second weakness. We appreciate your insightful feedback, which highlights the need to better contextualize our findings within the broader debate of associative versus propositional reasoning in cognitive science. In our paper, we explore how LLMs, particularly those based on transformer architectures, serve as a unique framework that integrates both reasoning paradigms.
>
> LLMs are traditionally viewed as statistical machines, predominantly leveraging associative mechanisms to generate predictions based on patterns learned from vast datasets. This perspective aligns with associative models, suggesting that cognitive processes operate by linking co-occurring elements within an experience. However, these models also exhibit emergent properties that resonate with propositional reasoning—namely, the ability to reason over multi-hop problems. Our work demonstrates that the ability to perform the sequential task of 2-hop problems is also accompanied by the association to the intermediate solutions stressing (again) the associative nature of the model.
>
> Note that unlike previous suggestions like ACT-R that explicitly incorporate propositional and associative elements, LLMs are "statistical machines" that demonstrate the capability to perform sequential operations that are also accompanied by associations. Notably, this characteristic of LLMs is an emergent property of the system, due to the fact that there were no explicit constraints in the training process designed to create this behavior.
>
>
> Question 1:
>
> Our formulation may generalize to other multi-hop tasks, like comparisons, but adjustments are likely needed. For example: “Which color name is longer - banana or apple?” Future work should explore where in the model the activations for “yellow” and “red” occur and how the comparison is made. This topic is important for the field of implicit reasoning and deserves its own research paper.
> Regarding n-hop queries with more than 2 steps: We believe that our formulation can handle any number of reasoning steps by adding more linear transformations to the computation. In practice, it appears that LLMs struggle to perform more than two implicit reasoning steps (and even the two-step case is challenging). Future work might explore how to leverage our findings in order to teach models how to solve more complicated queries.
>
> Question 2:
>
> We thank the reviewer for that question. The way we see it, the term “use it” in this context is referring to the causality of the process. We agree that understanding the causes behind model predictions is crucial. We believe that developing methods for causally interpreting LLMs is essential, but it remains an open area of research [1,2,3].
>
> However, while we do not claim causality, our results show that our linear model can explain a high percentage of the variance in the data, even though it was calculated on unseen data (using the k-fold method). This finding highly valuable, regardless of whether the process has a causal nature.
>
> We also want to highlight that the output embedding, which represents the final answers, was created by performing a series of computations on the embedding from the middle layer, which represented the intermediate answers. A non-causal process means that the LLM layers manipulate the middle embedding in a way that is invariant to the coordinates holding the representations of the intermediate answers. The output of this invariant process creates a vector that is surprisingly linearly dependent on the exact coordinates, which were not part of the process. We find this alternative hypothesis unlikely, but we did not prove it to be false.
>
> Lastly, following your question, we added a new section to the appendix. Please refer to section C.5 in the latest version of our paper. There, we present an analysis showing that when our linear model didn't capture most of the variance in the process, the LLM also failed to answer the reasoning questions. This analysis can provide some sense of the extent to which distributional reasoning is an essential part of the reasoning process.
>
> [1] Causal Abstractions of Neural Networks. https://arxiv.org/abs/2106.02997
>
> [2] CausaLM. https://arxiv.org/abs/2005.13407
>
> [3] Causal Inference in Natural Language Processing.
> https://arxiv.org/abs/2109.00725

---

> ### Comment · Reviewer_b2q5 · 2024-11-25
> **Response by R.b2q5**
>
> **1.** I am confused by some of the constraints that the authors have placed on the datasets for which the proposed analysis method can work. You're claiming that the linear transformation-based model can only work if the questions are a specific kind of unnatural:
>
> > In addition, the questions should be built in a way that the final answer shouldn't be related to the question subject directly ('y' is not related directly to 'banana' without the context of its color).
>
> Why does the method not work for more natural multi-hop questions whose sub-questions are more topically consistent?
>
> **2.** I appreciate that the authors made an effort to add another dataset to the paper's experiments, albeit one that is very simple and subject to the same constraints that I mentioned on the previous question.
>
> However, if the experiment with this new dataset is going to be included in the paper, it should not be relegated to a tiny paragraph in the appendix. The authors should better explain how the new dataset was created beyond saying "we used Gemini".
>
> **3.** I appreciate that the authors acknowledge in their rebuttal that their analysis does not indicate causality between the model predictions and the suggest-than-narrow-down mechanism captured by the linear models, and therefore it is perhaps an overclaim to say that the model necessarily "uses" this process on an instance-by-instance basis. I agree that the analysis is still valuable regardless of whether you have shown causality, but the current paper draft is littered with references to LLMs "using" this process, which is an overclaim:
>
>
> L24  "Our findings can help uncover the strategies that LLMs use to solve reasoning tasks"
>
> L64 "what strategy does the model use when applying the implicit approach?"
>
> L116 "we conduct two experiments that show that LLMs use the same reasoning process even when they hallucinate their answers."
>
> L507  "Evaluating this consistency can help assess the ability of LLMs to use valid reasoning processes"
>
>
> I appreciate the note at the end of the limitations section that "although the statistical analyses in this paper are quite convincing, they do not show direct causality", but the main body of the paper should be clearer, particularly in the introduction, about whether the findings indicate a strong associates vs evidence of usage.

---

> ### Author Response · Authors · 2024-11-30
>
> We thank the reviewer for their detailed response. In addition to our reply below, we encourage the reviewer to refer to Section D of our paper, which was added following the rebuttal. We believe the demonstrations in this section will be helpful for observing our proposed mechanism in action.
>
> 1.
>
> We apologize for the confusion made by our response regarding the constraints of the dataset. The constraint the reviewer mentioned is actually intended to make the reasoning questions **harder** by requiring the model to use genuine reasoning strategies. If we won’t try to build the questions that way, there are two concerns that we will have to address:
>
> First, if the question is too natural, we might suspect that the model has seen it (or similar questions) in its training set, making it a case of memorization or shortcut learning [1]. The Compositional Celebrities dataset (and many other reasoning datasets) is designed to minimize the chances that such questions appeared in training.
>
> The second concern is that poorly designed questions may allow shortcuts that bypass the first reasoning step. For example: "What is the last name of the son of the president of the US?". Because the answer (Biden) is strongly associated with the subject (Joe Biden), the model might not go through the intermediate answer.
>
> These two concerns are the reason for many reasoning datasets to combine unlikely pairs of concepts, such as a celebrity's birthplace and its Russian name, which rarely appear together in natural language questions. If that wasn’t the case, there might not be any reasoning steps to follow. Distributional reasoning may still be a significant component in answering these types of questions, but the results are likely to be noisier due to the lack of true reasoning—making such questions less interesting to explore.
>
> 2.
>
> We certainly agree with the reviewer and will add the prompts used to create the dataset. It's worth noting that these prompts were very straightforward (e.g., "Please create a list of 100 bird names"). Moreover, the full list of 1593 prompts will be provided in the supplementary materials. Additionally, there will be references to the new results in the main body of the paper, explaining that these were added to expand the results and show more diversity in the relations. We may also add more relations for the camera-ready version.
>
> Lastly, for more clarity, the constraints behind the choice of datasets (as presented in the global rebuttal) will be integrated into Section 3.3.
>
> 3.
>
> We appreciate the author highlighting the need for clarifying this part. The research question presented in our paper is a question of the actual process within the model, which is essentially a question about the causal nature of the process. While causality in LLMs is still an open area of research, most studies in our field aim to gather supporting evidence. As mentioned in our previous comment, we see the non-causal alternative hypothesis to our findings unlikely, but we didn’t prove it.
>
> When presenting our proposal in the introduction, we stated: “Our proposal… is demonstrated by showing that activations of potential final answers in the output layer can be approximated using a linear model…”. Later, we wrote: "Without testing direct causality, we demonstrate a strong relation between the distributions…”. In addition, we changed some of the lines the reviewer mentioned:
>
> L24: "Our findings propose a new strategy that LLMs might use to solve reasoning tasks."
>
> L116: “We conducted two experiments that demonstrate our findings also apply in cases of hallucinations, suggesting that the same reasoning processes occur even when LLMs generate fabricated answers”
>
> L507: “Developing methods to assess this consistency, ideally by identifying its causal mechanisms, can pave the way for evaluating the validity of reasoning processes in LLMs—an aspect that may be more important than the output itself.”
>
> Regarding L64: In this line we are presenting the research question, which is what the model actually using. It is worth noting that it is not unusual to present the research question in that form - both papers referenced in that paragraph [2,3] are presenting their research question similarly (”Do LLMs retrieve factual information stored in their parameters and perform latent multi-hop reasoning..?” [2]). Therefore, we prefer to keep this line without changes.

---

> > ### Author Response · Authors · 2024-11-30
> >
> > [1] Robert Geirhos, Jorn-Henrik Jacobsen, Claudio Michaelis, Richard Zemel, Wieland Brendel, Matthias Bethge, and Felix A Wichmann. Shortcut learning in deep neural networks. Nature Machine Intelligence, 2(11):665–673, 2020.
> >
> > [2] Sohee Yang, Elena Gribovskaya, Nora Kassner, Mor Geva, and Sebastian Riedel. Do large language models latently perform multi-hop reasoning? arXiv preprint arXiv:2402.16837, 2024.
> >
> > [3] Zhaoyi Li, Gangwei Jiang, Hong Xie, Linqi Song, Defu Lian, and Ying Wei. Understanding and patching compositional reasoning in llms, 2024.

---

### Official Review · Reviewer_F2fn · 2024-11-04

**Soundness:** 3
**Presentation:** 3
**Contribution:** 3
**Rating:** 5
**Confidence:** 4

**Summary:**

This paper introduces a method to analyze multi-hop reasoning in LLMs by modeling their prediction process as a linear transformation between semantic spaces. The authors find that intermediate model layers create interpretable embeddings representing potential answers, activating specific token subsets and suggesting parallel reasoning paths. This framework offers insights into how LLMs mimic thought processes, even without direct task knowledge, with implications for understanding AI reasoning and cognitive modeling.

**Strengths:**

- This paper addresses a critical challenge in understanding LLM reasoning by probing the multi-hop reasoning processes. The discussion on human cognitive processes and LLM reasoning mechanisms is also inspiring.

- The study presents a clear illustration of distributional reasoning in LLMs (esp. with the example in Figure 2), which is a novel finding to my knowledge.

**Weaknesses:**

- While the illustration of distributional reasoning is interesting, the remaining analysis seems confusing to me. For example, what's the motivation of the analysis that uses the matrix to approximate the second-hop reasoning? The paper mentioned "This method reveals the extent to which the second-hop operation is invariant to the prompt specifics." why is this an important question to explore? and what's the takeaway from this analysis?

- The writing needs to be improved to make the logic clearer. Especially, there should be strong and clear motivation before going into any notations and technical details of the analysis. The current version makes readers very easy to get lost.

**Questions:**

See the weakness.

---

> ### Author Response · Authors · 2024-11-18
>
> We appreciate the reviewer's comments and will address the mentioned issues to the best of our ability.
>
> Regarding the first weakness, we appreciate this comment as we believe this finding is an important part of our paper. We will make its motivation clearer following your feedback. The matrix analysis is an important part of our main claim because it demonstrates how the reasoning process works. Let us take as an example the question from the paper: "What is the first letter of the name of the color of a common banana?". After obtaining the color(s), it doesn't matter anymore what fruit the question was about. Meaning, there is a unique function, which in our case is approximately linear, that will make the same transition from "yellow" to "y" even if the question was about the color of a lemon. The fact that our matrices can generalize to new prompts demonstrates how the reasoning process acts as a two-step pipeline—like we expect humans to solve it, and like we expect the Chain of Thoughts method to solve it.
>
> As far as we believe, this is more than just a nuance, and it is a main part of the novelty of this work. This reveals an emergent ability of LLMs for general multi-step task-solving mechanisms. Moreover, we claim that this mechanism is a stronger demand than just solving the tasks correctly, and future work will need to examine how to add this evaluation as a constraint to the training process.
>
> Regarding the second weakness, we are very sorry for the confusion caused, and we are working on improving the clarity of our paper. Beyond adding a stronger motivation paragraph to the matrix analysis, is there any other specific part that you find confusing?

---

> > ### Comment · Reviewer_F2fn · 2024-11-22
> >
> > If I understand correctly, your claim is, the second-hop reasoning can be done solely based on an intermediate representation (in your case, "yellow"), while irrelevant to previous contexts?
> >
> > I still don't understand why this can explain "how the reasoning process works" (in LLMs). This seems to be **an attribute decided by the problem structure**. You could also find other questions, where the second reasoning steps relies on more than one premise. E.g., for a question like "1 + 2 + 4 = ?", you may get an intermediate representation of "3", but the second step still relies on the question.

---

> ### Author Response · Authors · 2024-11-24
>
> Yes, that is the claim. Our results were calculated using k-fold validation, showing that the linear transformation obtained from 80% of the prompts generalized to the remaining prompts. This demonstrates that the second step remains approximately constant within questions of the same relation type.
>
> If we understand the reviewer's concern correctly, they worry that the activation of "yellow" is merely an artifact of the prompt structure rather than a reflection of the actual reasoning process.
>
> In our view, our results show something more convincing than that. If we look at Figure 2.b as an example, we can see that not only is "yellow" activated, but other colors are as well. While this could be just an artifact of the question, a surprising pattern emerges—the order "yellow," "brown," "green," "red," "white" matches exactly with the order of their corresponding first letters "y", "b", "g", "r", "w”. We find it difficult to explain how these two activation patterns could not be connected to each other.
>
> Additionally, we believe that the hallucination experiments (section 4.3) demonstrate our hypothesis in a much more challenging setting and provide important evidence for our proposed mechanism.
>
> If we want to delve deeper into whether the selected activations truly cause the activations in the output, we're entering the realm of causality. We agree that understanding the causes behind model predictions is crucial. We believe that developing methods for causally interpreting LLMs is essential, but it remains an open area of research [1,2,3]. While we do not claim causality, we believe that our findings are valuable regardless of whether the process has a causal nature.
>
> However, we want to highlight that the output embedding, which represents the final answers, was created by performing a series of computations on the embedding from the middle layer, which represented the intermediate answers. A non-causal process means that the LLM layers manipulate the middle embedding in a way that is invariant to the coordinates holding the representations of the intermediate answers. The output of this invariant process creates a vector that is surprisingly linearly dependent on the exact coordinates, which were not part of the process. We find this alternative hypothesis unlikely, but we did not prove it to be false.
>
> Additionally, please refer to sections C.5 and D, which we added to the paper after this rebuttal. We believe they can provide more intuition about the dominance of our proposed mechanism in LLM reasoning.
>
> [1] Causal Abstractions of Neural Networks. https://arxiv.org/abs/2106.02997
>
> [2] CausaLM. https://arxiv.org/abs/2005.13407
>
> [3] Causal Inference in Natural Language Processing. https://arxiv.org/abs/2109.00725

---

> > ### Comment · Reviewer_F2fn · 2024-11-25
> >
> > > If we understand the reviewer's concern correctly, they worry that the activation of "yellow" is merely an artifact of the prompt structure rather than a reflection of the actual reasoning process.
> >
> > Actually, that's not my concern. I agree that the model uses "yellow" as an intermediate representation of the reasoning process, and I think the results in Figure 2 have clearly illustrated that (as mentioned in my review). What I don't understand is the role of your analysis regarding to matrix approximation of the second-hop.

---

> > > ### Author Response · Authors · 2024-11-25
> > >
> > > We apologize for the confusion and will do our best to answer your question while clarifying our motivation for this analysis in the paper.
> > >
> > > According to our formulation (section 3.2), each relation has an associated (linear) function Q2 that performs the second operation identically on every prompt with that relation. We use regression analysis to identify this Q2 matrix for each relation.
> > >
> > > This analysis, using the k-fold method, directly supports our claim: a function trained on certain prompts can generalize to new ones. This means the second operation remains identical across prompts (otherwise it wouldn't have generalized that well). It is not just that “yellow” is been used in both banana and lemon questions. It assures that the operation that converts “yellow” to “y” and “green” to “g” is the same operation across different items in different colors. Showing only the activation patterns (as in figure 2) is not convincing enough for demonstrating our mechanism.
> > >
> > > Moreover, while the colors-letters example effectively demonstrated our point, some relations are more complex. Take the callingcode relation, which maps 117 country names to just 9 digits. In this case, **each** country contributes differently to the activation of **each** digit. The matrix captures this complex operation, and allow us to present this operation as a linear transformation between two semantic groups.
> > >
> > > Finally, the matrix analysis reveals that the operation is approximately linear. While this linearity is secondary to our main claim about the reasoning process, we consider it a significant finding in terms of LLM interpretability.

---

### Author Response · Authors · 2024-11-19

We sincerely appreciate the comprehensive feedback provided by the reviewers. We are diligently working to enhance our paper in accordance with your comments, and we aim to address each point thoroughly and effectively.

Thanks to the reviewers' insightful comments, we've expanded the paper with three new sections in the appendix. For the camera-ready submission, these sections will be integrated into the main body of the paper. Please refer to sections C.4, C.5, and D in the latest version for these additions.

In response to questions about our dataset selection, we would like to provide further explanation for our choices:

In this work, we are trying to uncover a general mechanism of LLMs implicit reasoning. This mechanism contains two main characteristics. One, LLMs activate parallel reasoning paths, and second, the process works as a pipeline of two sequential stages in which the second relies only on the output of the first (’y’ will be the first letter of ‘yellow’ no matter if the question was about ‘banana’ or ‘lemon’). When trying to demonstrate this mechanism, there are several constraints that we need to keep in mind:

First, LLMs are not very good at solving multi-step questions implicitly. This is the reason why the Chain-of-Thought method has become such a popular approach. This fact limits us to keeping the structure of the prompts relatively simple, because otherwise they just won't be able to reach the correct answer, and no reasoning process could be traced. We hope our paper will promote directions for addressing this problem in future models.

Second, as in other scientific disciplines, our proposed approach for demonstrating the proposed mechanism requires its own limitation of data curation. This doesn't mean that our results don't apply to other structures and relations, but other relations will need a different approach to examine them. Same as brain mechanisms discovered in fMRI experiments might not apply exactly to any other situation outside of the lab, the experimental design constrains its setting, yet it is still highly valuable.

Now in more details about our choice of dataset:

Our framework describes the reasoning process as a transformation between two semantic categories. In order to conduct the analyses we proposed, we need a dataset of two-step questions with unique characteristics: each question should contain sub-questions which relate to two distinctive semantic groups (e.g., colors, letters, cities, etc.). In addition, the questions should be built in a way that the final answer shouldn't be related to the question subject directly ('y' is not related directly to 'banana' without the context of its color). Lastly, the questions should be constructed in a way that the answer could be generated in a single token.

The Compositional Celebrities dataset is ideal for demonstrating our claims because the alternative intermediate and final answers for its questions come from relatively distinct categories. The dataset's semantic categories (countries, digits, domains, etc.) are well defined, making it easy to identify possible alternatives answers in the model’s logits. The dataset has 6547 prompts in 14 relation types. Each type has 468 prompts, allowing us to effectively fit the linear models.

These data properties are rare in other datasets, often lacking diverse relation types or clear semantic categories. For instance, the dataset in [1] has only 1000 prompts split into 19 relation types, but with only 3 having over 100 prompts. All relations are attributes of a person (e.g., "The place of birth of the father of John Penn"). "Father of", for example, isn't a well-defined category for our purposes, as it will be hard to obtain a vector representing all fathers. It doesn't mean the model doesn't use distributional reasoning. It may still use the activation of alternative options for John Penn's father to create the output result.

To expand our dataset and empirical results, we curated a synthetic dataset that allows us to track, in a controlled setup, the effects of intermediate activations on the output tokens. The hallucination experiments, which demonstrate our hypothesis in a much harder setting, use a novel type of dataset to assess internal semantic processes in a neutral manner.

Lastly, following the comments from this rebuttal, we added to the appendix results for 1593 new prompts, which are not questions regarding birthplaces of celebrities. These results will be integrated into the main paper by the camera-ready version. See C.4 section in the last revision.

[1] Memory Injections. https://aclanthology.org/2023.blackboxnlp-1.26/

---

### Author Response · Authors · 2024-12-03

We appreciate the detailed feedback provided by the reviewers. Your comments greatly improved our work.

We would like to highlight several novel contributions of this work, which we will clarify in the camera-ready version:

1. An analysis of multi-hop reasoning processes reveals the presence of parallel reasoning paths. This insight of multiple paths suggest a more precise understanding of the internal workings of LLMs.

2. A linear model fitted to each type of second-hop operation is used to predict unseen data, revealing the extent to which the second-hop operation is invariant. This indicates that the linear transformation performing the reasoning process remains the same across different prompts. For instance, a linear model fitted on “the first letter of the name of the color of a banana is” will predict the outcome of the same reasoning operation on an apple's color. This separates reasoning from the prompt specifics, which is crucial since questions about color names' first letters aren't inherently related to the fruits.

3. Our "hallucinations experiments" provide a new method for evaluating internal processes in LLMs. These experiments highlight the difference between expecting accurate answers from LLMs and expecting a valid reasoning process, which can sometimes produce incorrect answers. Moreover, the synthetic dataset we used, consisting of fictitious items, demonstrates a novel way of creating datasets for assessing the inner workings of LLMs.

4. Our analysis indicates that this parallel process is interpretable. Projecting the embedding to the vocabulary space, it is as if the model "thinks out loud" allowing us to trace its simultaneous reasoning processes. This finding is significant for interpretability and can help trace associative errors that lead to incorrect predictions.

5. This work examines the internal thinking processes of LLMs from a cognitive science perspective, raising fundamental questions about the nature of intelligence and cognition. Our observational findings reveal mechanisms that can be viewed as a mix of two main and competing approaches to cognitive modeling. This highlights the significance of exploring LLMs' inner workings for cognitive literature.

---

### Meta-Review · Area_Chair_1d8w · 2024-12-20

**Metareview:**

The paper investigates the internal multi-hop reasoning processes in large language models (LLMs). The authors propose that LLMs utilize a "distributional reasoning mechanism", whereby intermediate layers generate embeddings representing potential intermediate answers, which are then refined in subsequent layers. The study uses a synthetic dataset to demonstrate that these processes can be modeled using linear transformations between semantic spaces, suggesting the presence of parallel reasoning paths.

Strengths:
* Addresses a core problem in LLM research: understanding the nature of "reasoning" in LLMs.
* A novel methodological approach to tackle this problem: an interpretable analysis of internal multi-hop reasoning processes.
* A solid, clear, well-motivated experimental design.

Weaknesses:
* The paper relies on a single synthetic dataset and only two-hop questions, which raises questions about the generalizability of the findings to more complex or natural datasets. In its current form, the findings are overly narrow.
* The paper might suffer from confirmation bias / non-falsifiable hypothesis testing. The experimental results are consistent with but stop short of proving the main hypothesis (distributional reasoning). As noted by reviewer nbdn, corroborating evidence could be provided by experiments that rule out other alternative hypotheses or that show that certain interventions motivated by the hypothesis result in changes consistent with it. But it its current form, saying that the distributional reasoning is "demonstrated" in this paper does feel overblown.

Overall, I think this is a good but borderline paper, that provides findings that are likely to be of interest to the community but whose validity and robustness is still unclear.

**Additional Comments On Reviewer Discussion:**

This paper prompted a very lively discussion between authors and reviewers. During the rebuttal period, several key points were raised and discussed:

* **Validity, Generalizability, Robustness of Findings**: Reviewers ndbn and b2q5 raised concerns about the validity and generalizability of the results and all reviewers cited the limitation to a single dataset and two-hop questions as a major weakness. The authors responded by adding new sections and datasets to address these concerns, but the reviewers remained unconvinced.
* **Clarity and Motivation**: Reviewer F2fn found parts of the analysis confusing and lacking clear motivation. The authors attempted to clarify these points in their rebuttal, but the reviewer still found the explanations insufficient.
* **Dataset and Experimental Design**: Reviewer P5is appreciated the aditional datasets provided in the rebuttal but maintained concerns about the robustness of the findings across different datasets and more complex tasks.

---

### Decision · Program_Chairs · 2025-01-22

Reject